# An immune receptor complex evolved in soybean to perceive a polymorphic bacterial flagellin

Yali Wei[1,2], Alexandra Balaceanu[3], Jose S. Rufian [1], Cécile Segonzac [4], Achen Zhao[1,2], Rafael J. L. Morcillo [1] & Alberto P. Macho [1✉]

In both animals and plants, the perception of bacterial flagella by immune receptors elicits the activation of defence responses. Most plants are able to perceive the highly conserved epitope flg22 from flagellin, the main flagellar protein, from most bacterial species. However, flagellin from *Ralstonia solanacearum*, the causal agent of the bacterial wilt disease, presents a polymorphic flg22 sequence (flg22[Rso]) that avoids perception by all plants studied to date. In this work, we show that soybean has developed polymorphic versions of the flg22 receptors that are able to perceive flg22[Rso]. Furthermore, we identify key residues responsible for both the evasion of perception by flg22[Rso] in Arabidopsis and the gain of perception by the soybean receptors. Heterologous expression of the soybean flg22 receptors in susceptible plant species, such as tomato, enhances resistance to bacterial wilt disease, demonstrating the potential of these receptors to enhance disease resistance in crop plants.

[1] Shanghai Center for Plant Stress Biology, CAS Center for Excellence in Molecular Plant Sciences; Shanghai Institutes of Biological Sciences, Chinese Academy of Sciences, Shanghai 201602, China. [2] University of Chinese Academy of Sciences, Beijing 100049, China. [3] Institute for Research in Biomedicine (IRB Barcelona), The Barcelona Institute of Science and Technology (BIST), 08028 Barcelona, Spain. [4] Department of Plant Science, Plant Genomics and Breeding Institute, Research Institute of Agriculture and Life Sciences, Seoul National University, Seoul 08826, Republic of Korea. ✉email: alberto.macho@icloud.com

I n order to secure a stable food supply for the increasing world population, it is imperative to minimize crop losses due to environmental stresses, such as diseases caused by pathogens and pests. Plants lack an adaptive immune system with specialized immune cells; therefore, to fend off biotic threats, most plant cells should be able to perceive non-self signals, process them, and respond accordingly. Certain conserved microbial molecules are detected at the surface of plant cells as elicitors of immunity, and are generally termed pathogen-associated molecular patterns (PAMPs). The surface receptors that mediate such perception are called pattern-recognition receptors (PRRs). The subsequent immune responses triggered by PRR activation ultimately hinder pathogen proliferation, in a phenomenon called pattern-triggered immunity (PTI)[1]. Given the importance of the flagellum for bacterial lifestyle, the abundance of flagellin (the major protein that builds the flagellum), and the conservation of certain peptides in its sequence, flagellin constitutes an excellent tell-tale molecule for animals and plants to detect the presence of a potential bacterial pathogen. As a consequence, most plants are able to perceive conserved epitopes of bacterial flagellin, such as the 22-amino acid peptide flg22[2]. Flg22 is perceived by a receptor complex formed by two trans-membrane co-receptors, FLAGELLIN-SENSITIVE 2 (FLS2) and BRASSINOSTEROID-INSENSITIVE 1-ASSOCIATED KINASE (BAK1), both of which contain extracellular leucine-rich repeats (LRRs) and an intracellular kinase domain[3,4]. The analysis of the crystal structure of Arabidopsis FLS2 and BAK1 ectodomains in complex with flg22 revealed that FLS2 LRRs bind specific residues of flg22, while BAK1 recognizes the C-terminus of FLS2-bound flg22[5]. The physical interaction between FLS2 and BAK1 leads to the activation of their intracellular kinase domains and the initiation of immune signalling[6].

*Ralstonia solanacearum*, the causal agent of the bacterial wilt disease, is a soil-borne bacterial pathogen able to infect more than 250 plant species[7,8]. Upon plant invasion through the roots, *R. solanacearum* reaches the vascular system and colonizes the whole plant;[9] subsequent bacterial replication and vascular clogging lead to plant wilting and death[10]. *R. solanacearum* is exceptionally resilient in environmental systems and extremely destructive, leading to enormous losses in crop production worldwide[8]. Interestingly, *R. solanacearum* is one of the few pathogens that have evolved polymorphisms in the flg22 sequence, avoiding perception by all plants tested so far, including Arabidopsis[11] and several crop plants from the *Solanaceae* family[12,13]. Similar cases include the modified flg22 sequences from *Pseudomonas cannabina pv. alisalensis* ES4326 (formerly known as *P. syringae pv. maculicola*), or *Agrobacterium tumefaciens*[2,14]. These and other examples suggest that allelic diversification in PAMPs represents a suitable virulence strategy for pathogens to avoid perception by plants[15]. Polymorphisms in *R. solanacearum* flg22 (flg22$^{Rso}$) abolish the recognition by FLS2/BAK1[5,12] (Supplementary Fig. 1a), but do not affect the function of the flagellum in bacterial motility. Previous attempts to perform targeted mutagenesis in FLS2 based on the analysis of the primary amino-acid sequence failed to confer responsiveness to flg22$^{Rso}$[16]. This suggests that, in order to generate gain-of-perception of flg22$^{Rso}$ by plant PRRs, we need to acquire a deeper understanding of which polymorphisms enable flg22$^{Rso}$ to avoid perception and how. In this work, we show that soybean has developed polymorphic versions of the flg22 receptors that are able to perceive flg22$^{Rso}$, revealing a dynamic co-evolution in the perception of conserved bacterial elicitors by plant immune receptors. Furthermore, we identify key residues responsible for both the evasion of perception by flg22$^{Rso}$ in Arabidopsis and the gain of perception by the soybean receptors. Heterologous expression of the soybean flg22 receptors in

susceptible plant species, such as tomato, enhances resistance to bacterial wilt disease, demonstrating the potential of these receptors to generate disease resistance in crop plants.

## Results and discussion
**Analysis of polymorphisms in *R. solanacearum* flg22.** The crystal structure of Arabidopsis FLS2 and BAK1 ectodomains has been solved in complex with flg22 from *P. aeruginosa* (an opportunistic animal pathogen that can cause disease in plants)[5,17]. The flg22 sequence can be divided in a N-terminal region that interacts with FLS2, and a C-terminal region that interacts with both FLS2 and BAK1[5]. A glycine residue in the position 18 (G18) is present in most immune-eliciting flg22 sequences, including those from *P. aeruginosa* (flg22$^{Pae}$) or the notorious plant pathogen *P. syringae* (flg22$^{Psy}$) (Supplementary Fig. 1a), and is essential for the interaction of FLS2-bound flg22 with BAK1[5]. Accordingly, mutations in G18 reduce the elicitation of immune responses by flg22$^{Pae}$ in Arabidopsis[5]. Flg22 peptides from different *Pseudomonas* species display high similarity, including those from *P. aeruginosa* and *P. syringae*, which show identical sequence in the residues 9-22 (Supplementary Fig. 1a). Compared with flg22$^{Pae}$ or flg22$^{Psy}$, the sequence of flg22$^{Rso}$ contains 9 amino acid polymorphisms, mostly concentrated in the C-terminal region (Supplementary Fig. 1a). Mutations in the residues $^{18}$GLQ$^{20}$ of flg22$^{Pae}$ to $^{18}$AYA$^{20}$ (equivalent residues in flg22$^{Rso}$) abolish elicitation in Arabidopsis[12]. Considering these data, we tested whether performing the reciprocal mutation in flg22$^{Rso}$ (i.e. from $^{18}$AYA$^{20}$ to $^{18}$GLQ$^{20}$) is sufficient to elicit immunity in Arabidopsis. As readout of immune elicitor activity, we used the fast PAMP-induced burst of reactive oxygen species (ROS), which requires the formation and activation of the PRR complex[18]. As a positive control in functional assays, we used flg22$^{Psy}$. Eliciting flg22 peptides, such as flg22$^{Psy}$, but not flg22$^{Rso}$, trigger an FLS2-dependent ROS burst in Arabidopsis leaves (Fig. 1a). Interestingly, flg22$^{Rso}$-GLQ did not show elicitor activity (Fig. 1a), suggesting that other polymorphisms in flg22$^{Rso}$ also contribute to avoiding perception by AtFLS2/AtBAK1.

To predict the relative importance of the different flg22$^{Rso}$ polymorphisms, we modelled the structure of the ternary FLS2LRR/BAK1LRR/flg22 complex using the published structure of the Arabidopsis FLS2 and BAK1 LRRs together with flg22$^{Pae}$[5]. We first estimated changes of binding free energy caused by single amino acid changes at all the positions that show sequence variation between flg22$^{Pae}$ and flg22$^{Rso}$ (9 residues). Binding free energies were calculated separately for the first step of flg22 perception (FLS2-flg22) and for the second step of FLS2/flg22 interaction with BAK1 (FLS2/flg22-BAK1). For the first step, the mutations of I9 to V, Q20 to A, and I21 to A seem to have a strong impact on binding affinity, as indicated by their comparatively high values of binding free energy loss (Fig. 1b), with the mutation Q20A being particularly disruptive (Fig. 1b and Supplementary Fig. 1b and c). For the second step, in keeping with previously published results, a mutation in G18 to A is predicted to compromise binding affinity (Fig. 1b and Supplementary Fig. 1d and e), probably by causing steric clashes that attenuate the existing interactions with BAK1 residues[5]. Interestingly, the mutation of I21 to A shows an even stronger predicted impact on binding affinity (Fig. 1b). An analysis of the ternary complex structure around this region reveals hydrophobic interactions involving residues from all three binding partners that would be disrupted upon mutation of I21 to A (Fig. 1c and Supplementary Fig. 1f). Therefore, it is tempting to hypothesize that I21, together with BAK1-T58 (and other nearby residues in FLS2), forms a "hydrophobic patch" that enables the GLQ region to form important polar interactions (Fig. 1c and Supplementary

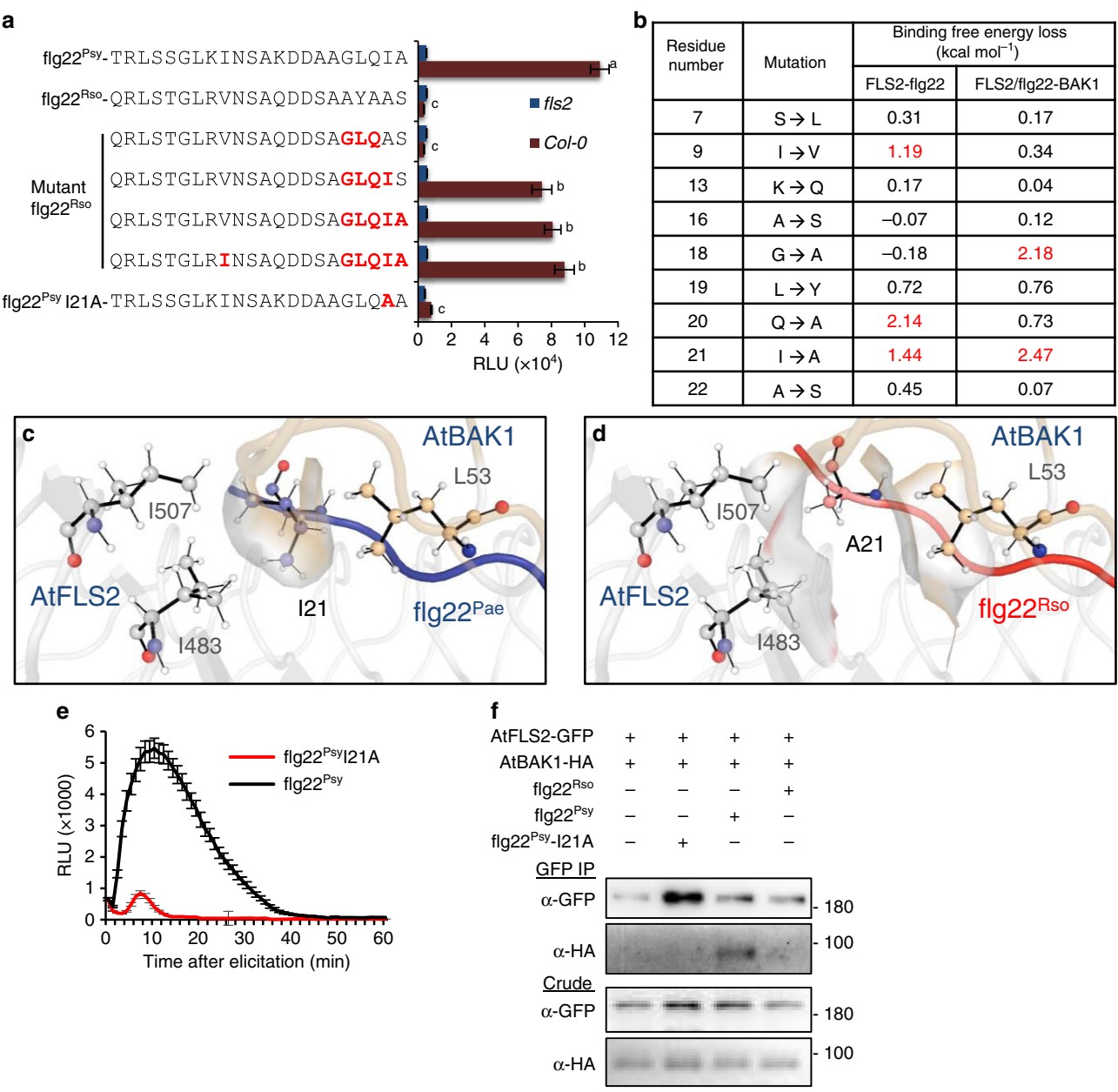

**Fig. 1 Analysis of polymorphisms in *R. solanacearum* flg22. a** ROS burst triggered by the indicated peptides (100 nM) in leaf discs from Arabidopsis Col-0 wild-type or an *fls2* mutant, measured in a luminol-based assay, and represented as accumulated relative luminescence units (RLU) (mean ± SEM, $n = 16$). The measurement was performed from 1 to 40 min after treatment. The elicitor peptide flg22$^{Psy}$ was used as control. Different letters indicate significant differences using a one-way ANOVA and Tukey correction for multiple comparisons ($p < 0.05$). This experiment was repeated 3 times with similar results. **b** Effect of single amino acid changes on binding free energy in a two-step triplex formation (FLS2-flg22 and FLS2/flg22-BAK1), measured in kcal mol$^{-1}$. Single amino acid changes at all positions that show sequence variation between flg22$^{Pae}$ and flg22$^{Rso}$ (9 residues) were simulated, and the change in binding free energy upon each individual mutation was estimated using the BeAtMuSiC program based on the structure of the protein–protein complex. **c** Graphical representation of the modelled interactions between flg22$^{Pae}$ and AtFLS2 around I21, showing the formation of a possible hydrophobic patch. **d** Alteration in the structure around I21 upon mutation of this residue to the A21 present in flg22$^{Rso}$. For **c**, **d** the change in solvent accessible cavities around these residues is shown in both cases. A different angle of this location is shown in Supplementary Fig. 1f and 1g. **e** Dynamics of ROS burst corresponding to the comparison between flg22$^{Psy}$ and flg22$^{Psy}$ I21A shown in (**a**). The measurement was performed from 1 to 60 min after treatment. **f** Co-immunoprecipitation assays to determine interactions between AtFLS2-GFP and AtBAK1-HA transiently expressed in *N. benthamiana* using *Agrobacterium tumefaciens*, upon treatment with the indicated peptides (1 μM). Peptide treatments were done 2 days after *A. tumefaciens* infiltration, and protein samples were taken 10 min after peptide treatments. Immunoblots were analysed with anti-GFP and anti-HA antibodies. Protein marker sizes (KDa) are provided for reference. This experiment was repeated three times with similar results.

Fig. 1f); the presence of an alanine would break these hydrophobic interactions, allowing solvent penetration that would compete for hydrogen bonding, shielding and weakening the interactions in the GLQ region (Fig. 1d and Supplementary Fig. 1g). Intriguingly, most R. solanacearum strains sequenced to date (118/155) have the same flg22 sequence, and all of them show the same polymorphisms as the predominant flg22$^{Rso}$ sequence in residues 9, 18, 19, and 20 (Supplementary Fig. 2); in the residue 21, a 16.7% of the strains presented an S, instead of the predominant A present in most strains (83.3%) (Supplementary Fig. 2).

Supporting the predicted importance of the I21A polymorphism, additional mutation of A21 to I in the flg22$^{Rso}$-GLQ mutant ($^{18}$GLQI$^{21}$) conferred elicitation activity in Arabidopsis (Fig. 1a). An additional mutation of the S22 to A in flg22$^{Rso}$-GLQI ($^{18}$GLQIA$^{22}$) did not enhance its elicitor activity (Fig. 1a). Since the polymorphism I9 to V was also predicted to impact binding activity (Fig. 1b), we added this mutation to the previously tested peptide (rendering flg22$^{Rso}$-I-GLQIA), and observed no further enhancement in its elicitor activity (Fig. 1a). Therefore, both prediction based on structural modelling and peptide elicitation assays raise the possibility that I21 is indeed a key residue to seal the pocket of interactions in the GLQ region. To further validate this notion, we mutated flg22$^{Psy}$ I21 to A, and observed that the single I21A mutation is sufficient to almost completely abolish the elicitation activity of this peptide (Figs. 1a and e). In agreement with its reduced ability to elicit ROS burst, the flg22$^{Psy}$-I21A mutant peptide did not trigger a detectable interaction between AtFLS2 and AtBAK1, analysed by co-immunoprecipitation (coIP) upon transient expression in Nicotiana benthamiana (Fig. 1f). Altogether, these data indicate that I21 is essential for flg22 perception, and that the I21A polymorphism may play a key role in the avoidance of perception in flg22$^{Rso}$.

**Soybean FLS2/BAK1 perceive R. solanacearum flg22.** Solanaceous plants have been reported as susceptible hosts for numerous R. solanacearum strains[19], and we previously found that several plant species from the Solanaceae family, including tomato, potato, pepper, tobacco, and N. benthamiana, cannot perceive flg22$^{Rso}$[13]. We decided to test the potential recognition of flg22$^{Rso}$ in plants from the Fabaceae family, which comprises several species susceptible to R. solanacearum[19,20]. Interestingly, while common bean (Phaseolus vulgaris), pea (Pisum sativum), peanut (Arachis hypogaea), and the model legume Medicago truncatula showed no responsiveness to flg22$^{Rso}$ (Fig. 2a and Supplementary Fig. 3), soybean (Glycine max) showed a clear robust ROS burst in response to flg22$^{Rso}$ (Fig. 2a and b). Soybean was also able to respond to treatment with recombinant flagellin from R. solanacearum (Fig. 2c), confirming that soybean has the ability to perceive R. solanacearum flagellin. The most obvious candidate PRRs to mediate this perception are the soybean FLS2/BAK1 orthologs. The sequenced genome of soybean Williams82 contains two FLS2 orthologs, named GmFLS2a and GmFLS2b[21], encoding proteins 54.47% and 53.69% identical to AtFLS2, respectively. Since N. benthamiana does not perceive flg22$^{Rso}$[13] and allows transient expression of heterologous genes, it constitutes an excellent system to analyse gain of perception of flg22$^{Rso}$. The sole expression of GmFLS2a in N. benthamiana from a 35S promoter did not confer responsiveness to flg22$^{Rso}$ (Fig. 2d). However, the simultaneous expression of GmFLS2a and GmBAK1 (encoding a protein 84.67% identical to AtBAK1), led to a clear ROS burst upon flg22$^{Rso}$ treatment (Fig. 2d). The expression of GmFLS2b alone did confer responsiveness to flg22$^{Rso}$ (Fig. 2d), probably acting together with the endogenous NbBAK1, although the intensity of the response was enhanced upon co-expression with GmBAK1 (Fig. 2d). ROS

dynamics in responsive tissues showed a single or double peak in ROS production around 5–20 min after elicitation, but single or double peaks were not reproducibly associated to any of the specific constructs used. The co-expression with GmBAK1 did not affect the accumulation of GmFLS2a/b significantly (Fig. 2e), and we consistently detected a lower accumulation of GmFLS2b, despite conferring stronger responsiveness, in comparison to GmFLS2a (Fig. 2d and e). Overexpression of AtFLS2 and AtBAK1 in N. benthamiana did not confer responsiveness to flg22$^{Rso}$ (Supplementary Fig. 4), indicating that the observed responsiveness is specific of the soybean PRRs. Consistent with the ROS results, flg22$^{Rso}$ triggered the association between GmFLS2b and GmBAK1 expressed in N. benthamiana, determined by coIP (Fig. 2f) and Förster resonance energy transfer—fluorescence lifetime imaging (FRET-FLIM) (Fig. 2g) assays. The activation of the soybean PRR complex by flg22$^{Rso}$ was further evidenced by downstream MAPK activation (Fig. 2f). Altogether, these results indicate that: (i) soybean is able to perceive flg22$^{Rso}$; (ii) expression of GmFLS2a and GmBAK1 in non-responsive plants confers responsiveness to flg22$^{Rso}$; (iii) expression of GmFLS2b in non-responsive plants is sufficient to confer responsiveness to flg22$^{Rso}$, but co-expression with GmBAK1 is required for full responsiveness; and (iv) flg22$^{Rso}$ triggers the interaction between GmFLS2b and GmBAK1 and the activation of downstream signalling. These findings reveal an exceptional case of dynamic co-evolution in the perception of conserved bacterial elicitors by plant PRRs.

**GmFLS2 polymorphisms are important for flg22$^{Rso}$ perception.** Flg22$^{Rso}$ evades recognition by most plants tested so far. Therefore, we reasoned that GmFLS2 and GmBAK1 must have evolved specific mutations in their extracellular domains to enable the perception of flg22$^{Rso}$. An alignment of the primary amino acid sequence of the extracellular domains of GmFLS2 and GmBAK1 with their Arabidopsis counterparts did not identify significant polymorphisms in key residues known to mediate interaction with flg22 residues that present relevant polymorphisms in flg22$^{Rso}$ (Supplementary Figs. 5 and 6). Therefore, we reasoned that polymorphisms in additional residues may alter the 3D structure of the binding interface in the GmFLS2/GmBAK1 complex to adapt to the polymorphisms in flg22$^{Rso}$, and employed homology-based structural modelling to identify potential sites responsible for the gain of recognition by GmFLS2/GmBAK1. Given its stronger responsiveness to flg22$^{Rso}$ (Fig. 2d), we used GmFLS2b for this analysis.

To set a basis for our homology modelling analysis, we determined key residues in AtFLS2/AtBAK1 important for binding to flg22$^{Pae}$ by alanine scanning, considering separately the two stages of the perception process (FLS2 + flg22 and FLS2/flg22 + BAK1). This analysis was interpreted to reveal hotspots in the structure where polymorphisms within the PRRs may impact flg22 perception (Supplementary Fig. 7). Then, we performed homology-based modelling of the structure of the extracellular domains of GmFLS2b and GmBAK1 bound to flg22$^{Rso}$, using the published AtFLS2/AtBAK1/flg22$^{Pae}$ structure[5] as template (Supplementary Fig. 8). Guided by the previously identified hotspots, we located important residues for binding of flg22 in AtFLS2 that show mutations or structural discrepancies between the AtFLS2/AtBAK1/flg22$^{Pae}$ and GmFLS2b/GmBAK1/flg22$^{Rso}$. We focused on three regions: GmFLS2 polymorphisms predicted to be associated with flg22$^{Rso}$Q13, with flg22$^{Rso}$AYA (18-20), and with flg22$^{Rso}$A21 (Fig. 3a–i). To determine the importance of these GmFLS2b polymorphisms for the perception of flg22$^{Rso}$, we performed site-directed mutagenesis in these residues, replacing them with the amino acid present in the equivalent position in

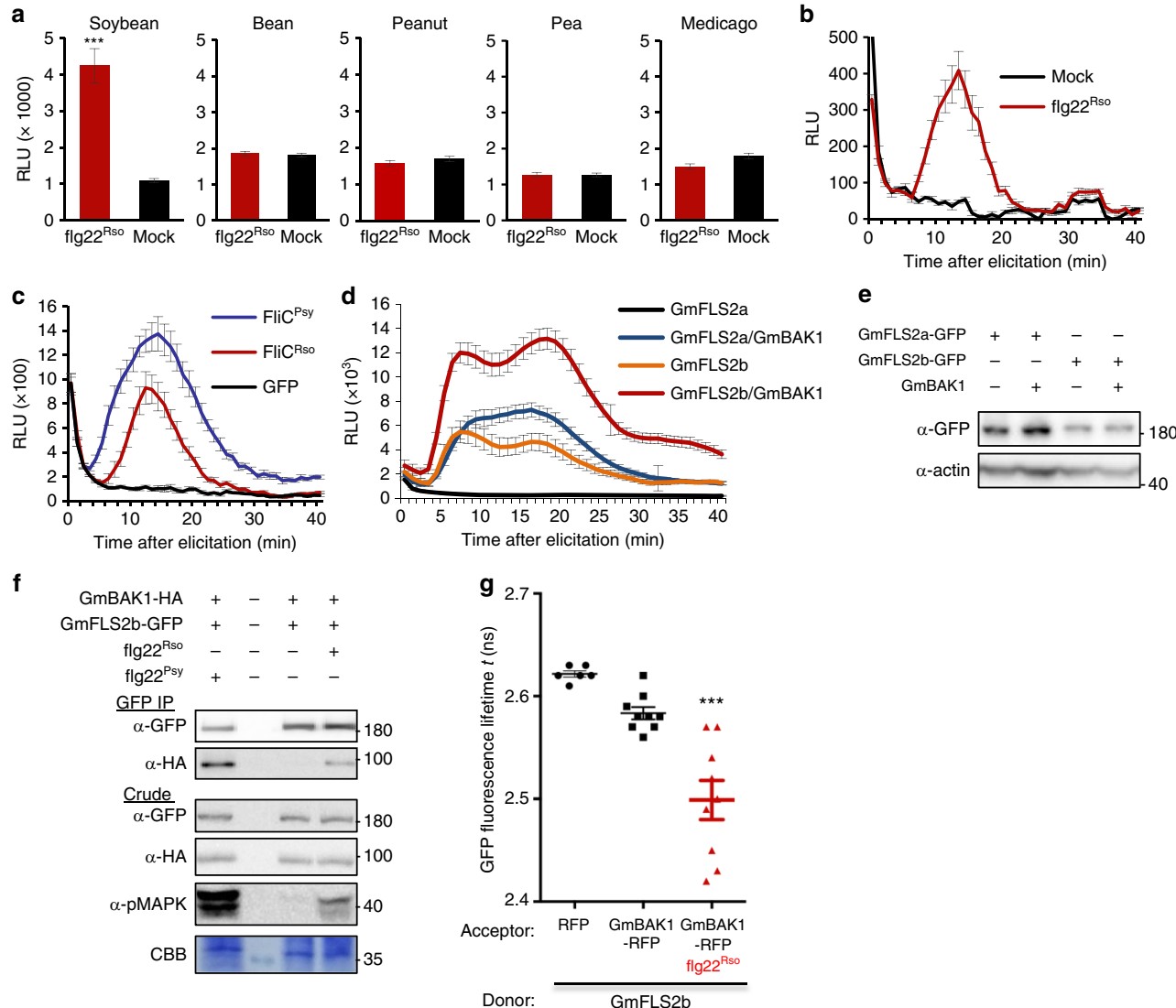

**Fig. 2 The soybean FLS/BAK1 complex perceives *R. solanacearum* flg22. a** ROS burst triggered by flg22[Rso] (100 nM) or a mock (water) treatment in leaf discs from the indicated plant species, measured in a luminol-based assay, and represented as accumulated relative luminescence units (RLU) (mean ± SEM, *n* = 8). Asterisks indicate significant differences compared to the mock control as established by a Student's *t* test (*p* < 0.001). The measurement was performed from 5 to 40 min after treatment to avoid the effect of the background signal during the first 5 min. **b** Dynamics of ROS burst triggered by flg22[Rso] (100 nM) or a mock (water) treatment in leaf discs from soybean. The measurement was performed from 1 to 60 min after treatment. **c** Dynamics of ROS burst triggered by 100 nM full-length recombinant flagellin (FliC) from *R. solanacearum* GMI1000 (FliC[Rso]) or *P. syringae* DC3000 (FliC[Psy]) in soybean leaf discs, measured in a luminol-based assay, and represented as relative luminescence units (RLU) (mean ± SEM, *n* = 8). Recombinant GFP protein was used as control. The measurement was performed from 1 to 40 min after treatment. **d** Dynamics of ROS burst triggered by flg22[Rso] (100 nM) in *N. benthamiana* leaf tissues expressing the indicated GmFLS2 versions (with a C-terminal GFP tag), alone or together with untagged GmBAK1, from a *35S* promoter. ROS was measured in a luminol-based assay, and represented as accumulated relative luminescence units (RLU) (mean ± SEM, *n* = 16). The measurement was performed from 1 to 40 min after treatment. **e** Accumulation of GmFLS2a/b-GFP expressed in (**d**). Immunoblots were analysed using anti-GFP and anti-actin (to verify equal loading). **f** Co-immunoprecipitation assays to determine interactions between GmFLS2-GFP and GmBAK1-HA transiently expressed in *N. benthamiana*, upon treatment with the indicated peptides (1 μM) for 10 min. Immunoblots were analysed with anti-GFP, anti-HA, and anti-pMAPK, and stained with Coomassie Brilliant Blue (CBB) to verify equal loading. In immunoblots, protein marker sizes (KDa) are provided for reference. **g** Interaction between GmFLS2-GFP and GmBAK1-RFP determined by FRET-FLIM upon transient co-expression in *N. benthamiana* leaves. Free RFP was used as a negative control. Lines represent average values (*n* = 8) and error bars represent standard error. Asterisks indicate significant differences compared to the RFP control as established by a Student's *t* test (*p* < 0.001). All the experiments were repeated at least three times with similar results.

AtFLS2. None of these mutations caused a significant alteration on GmFLS2b accumulation or subcellular localization (Supplementary Fig. 9). We then co-expressed these mutant versions of GmFLS2b with GmBAK1 in *N. benthamiana*, and quantified the responsiveness of the transformed tissue to flg22[Rso], measured as ROS production. Mutation of GmFLS2b-Q248 (associated to

flg22[Rso]Q13) to E did not have a significant impact on the responsiveness to flg22[Rso] (Fig. 3j). Importantly, mutation of GmFLS2b-Q368 (associated to the flg22[Rso]AYA region) to F reduced the responsiveness to flg22[Rso] by approximately 50% compared to wild-type (WT) GmFLS2b (Fig. 3k and Supplementary Fig. 9), indicating that this residue is important for the

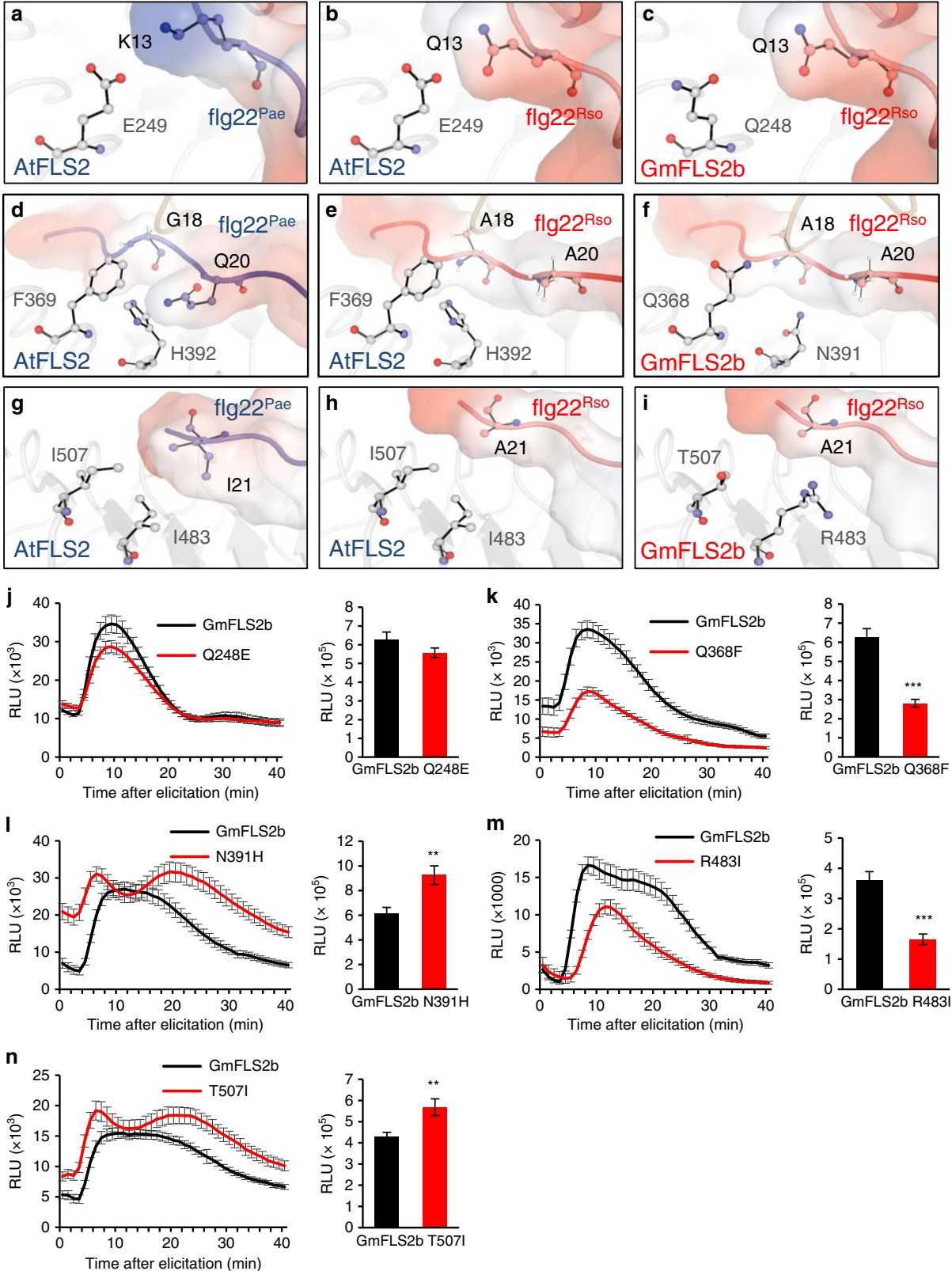

recognition of flg22$^{Rso}$ by GmFLS2b. On the contrary, mutation of GmFLS2-N391 (also associated to the flg22$^{Rso}$AYA region) to H did not reduce the responsiveness to flg22$^{Rso}$, but rather enhanced it slightly (Fig. 3l and Supplementary Fig. 9). Our previous data suggest that the polymorphism in the residue 21 of flg22$^{Rso}$ disrupts hydrophobic interactions between I21 and FLS2 residues (Fig. 1). Associated to this flg22 residue, GmFLS2b

presents the polar R483, instead of I483 in AtFLS2 (Fig. 3g–i). Mutation of GmFLS2-R483 to I reduced responsiveness to flg22$^{Rso}$ by approximately 40% (Fig. 3m and Supplementary Fig. 9), indicating that the polymorphism in this site is key for the recognition of flg22$^{Rso}$ by GmFLS2b. Surprisingly, a mutation in another polar polymorphic residue in this region, T507, to I, enhanced responsiveness to flg22$^{Rso}$ (Fig. 3n and Supplementary

**Fig. 3 Identification and functional analysis of GmFLS2/GmBAK1 residues important for perception of *R. solanacearum* flg22. a–i** Structural adaptions in GmFLS2b (compared with AtFLS2) for the recognition of flg22[Rso]. The analysis was focused around three regions: GmFLS2 polymorphisms predicted to be associated with flg22[Rso]Q13, with flg22[Rso]AYA (18-20), and with flg22[Rso]A21. **a**, **d**, **g** show orientations of key residues in AtFLS2 forming contacts that contribute significantly to the interaction with flg22[Pae] residues in each region. **b**, **e**, **h** show the effect of flg22[Rso] polymorphisms in the interaction with AtFLS2, disrupting contacts potentially important for complex formation. **c**, **f**, **i** show GmFLS2 polymorphisms at equivalent positions in the GmFLS2b/flg22[Rso] complex as potential structural adaptions responsible for gain of function. Note that the position of the residues Q248, Q368, N391, R483, and T507 in GmFLS2 is equivalent to that of the residues E249, F369, H392, I483, and I507, respectively, in AtFLS2, as shown in Supplementary Fig. 5. **j–n**. Dynamics of ROS burst triggered by flg22[Rso] (100 nM) in *N. benthamiana* leaf tissues expressing GmFLS2b or the indicated mutant versions (with a C-terminal GFP tag) together with untagged GmBAK1. ROS was measured in a luminol-based assay, and represented as accumulated relative luminescence units (RLU) (mean ± SEM, $n = 16$). The measurement was performed from 1 to 40 min after treatment. Bar charts show accumulated relative luminescence units (RLU) from 5 to 40 min after treatment to avoid the effect of the background signal during the first 5 min. Asterisks indicate significant differences compared to the control expressing GmFLS2b and GmBAK1 as established by a Student's $t$ test (**$p < 0.01$, ***$p < 0.001$). These experiments were performed 5–6 times, and a representative result is shown here. Composite data for all the replicates are shown in Supplementary Fig. 9.

Fig. 9). The reduced responsiveness of the Q368F and R483I mutants observed in ROS production assays was also validated, with similar results, using MAPK activation as readout (Supplementary Fig. 10). These data further confirm the importance of polymorphisms in and around the residue 21 of flg22 for perception by FLS2, and identify GmFLS2b-Q368 and R483 as important residues required for the gain of perception of flg22[Rso].

The structural modelling did not predict an obvious region in GmBAK1 required for the gain of perception of flg22[Rso]. The AtBAK1 residues 52–54 are involved in the interaction with flg22[Pae]-G18, and are conserved in the positions 54–57 of GmBAK1 (Supplementary Fig. 6). However, GmBAK1 contains a polymorphism in the nearby residue 58 (T to N) compared to AtBAK1 (Supplementary Fig. 6). Mutation of GmBAK1-N58 to T caused only a small reduction on responsiveness to flg22[Rso] (Supplementary Fig. 11). This result indicates that GmBAK1-N58 contributes to the gain of perception of flg22[Rso], but, together with our previous data (Fig. 2d), further suggests that polymorphisms in GmBAK1 make a minor contribution to the gain of perception of flg22[Rso].

**GmFLS2/GmBAK1 enhance resistance to *R. solanacearum*.** Although *R. solanacearum* can cause disease in more than 250 plant species within more than 50 families[19], and extensive surveys have been performed over the years to isolate *R. solanacearum* strains from diseased plants[20], to our knowledge, no strain has been isolated from soybean to date. This suggests that soybean displays natural resistance to *R. solanacearum*. To verify this in laboratory conditions, we inoculated soybean Williams82 with *R. solanacearum* (using the reference GMI1000 strain) by soil-drenching. While the same inoculum killed 92% of tomato plants in 14 days, none of the inoculated soybean plants showed any disease symptoms (Supplementary Fig. 12). Injection of a *R. solanacearum* suspension in the stem (an aggressive inoculation method used to bypass the root penetration process) killed 100% of tomato plants in 7 days, but only 27% of the inoculated soybean plants (Supplementary Fig. 12). These results support the idea that soybean is naturally resistant to *R. solanacearum*.

The transfer of PRRs between different plant species has been extensively used to confer additional responsiveness to pathogen elicitors, enhancing plant resistance to the corresponding pathogens[22–32]. Initial proof of the biological relevance of flg22 perception for resistance against bacterial pathogens came from the observation that flg22 treatment induced plant resistance against a subsequent bacterial inoculation[33]. To determine if heterologous expression of GmFLS2/GmBAK1 confers flg22[Rso]-induced resistance to *R. solanacearum*, we co-expressed both soybean PRRs in *N. benthamiana*, and performed leaf inoculation with *R. solanacearum* Y45, which is pathogenic in this species and is able to replicate rapidly upon infiltration in leaf tissues[34]. In

basal conditions, expression of GmFLS2/GmBAK1 caused a reproducible reduction in *R. solanacearum* multiplication, which was not statistically significant (Fig. 4a and Supplementary Fig. 13). Importantly, pre-treatment with flg22[Rso] strongly reduced *R. solanacearum* growth, to similar levels to those observed upon pre-treatment with flg22[Psy] (Fig. 4a), indicating that the GmFLS2/GmBAK1 receptor complex confers flg22-induced resistance to *R. solanacearum*. Induced resistance was also reflected by lower disease symptoms at later time points (Supplementary Fig. 13).

Tomato is an agriculturally important crop affected by *R. solanacearum*, and cannot perceive flg22[Rso][13]. Expression of GmFLS2/GmBAK1 in tomato roots conferred responsiveness to flg22[Rso] in root tissues, to similar levels to those observed upon treatment with flg22[Psy] (Fig. 4b), and rendered tomato plants more resistant to disease upon soil-drenching inoculation with *R. solanacearum* GMI1000 (Fig. 4c, d, and Supplementary Fig. 14), suggesting that the inter-family transfer of the soybean FLS2/BAK1 complex is a suitable strategy to enhance resistance to bacterial wilt in other crop plants.

**Natural PRR variants enhance disease resistance in plants.** Our work shows that the soybean GmFLS2/GmBAK1 has evolved polymorphisms that enable recognition of the polymorphic flagellin from *R. solanacearum*, which otherwise eludes perception in most plant species. Similarly, a recent report has shown that *Vitis riparia* has evolved an FLS2 allele that confers responsiveness to the polymorphic flg22 from *Agrobacterium tumefaciens*[35]. Traditional models consider that plants evolved to perceive PAMPs because of their importance for microbial housekeeping functions and the resulting evolutionary constriction in terms of their mutational ability. Our results and those reported by Furst et al.[35] demonstrate that not only PAMPs can mutate to elude plant perception, but certain plant species can also evolve to perceive polymorphic PAMPs, supporting the notion that PAMP perception is more flexible and evolutionarily dynamic than previously thought[15]. Interestingly, a recent report has shown that the *GmFLS2* alleles are required for the responsiveness to flg22[Psy] in soybean[21]. Our results also show that these receptors did not replace their ligand perception abilities, but rather broadened their recognition spectrum; although GmFLS2/GmBAK1 developed polymorphisms that allow the perception of flg22[Rso], they retain responsiveness to other flg22 sequences: flg22[Psy] treatment led to the interaction of GmFLS2b and GmBAK1 expressed in *N. benthamiana* (Fig. 2f), and the expression of both GmPRRs in tomato roots enhanced responsiveness to flg22[Psy] (Fig. 4b). Multiple studies have shown that the transgenic expression of new PRRs can confer recognition of PAMPs in plant species that were previously non-responsive (reviewed by Boutrot and Zipfel[36]). It is noteworthy that an extensive allelic diversification has been observed for certain plant

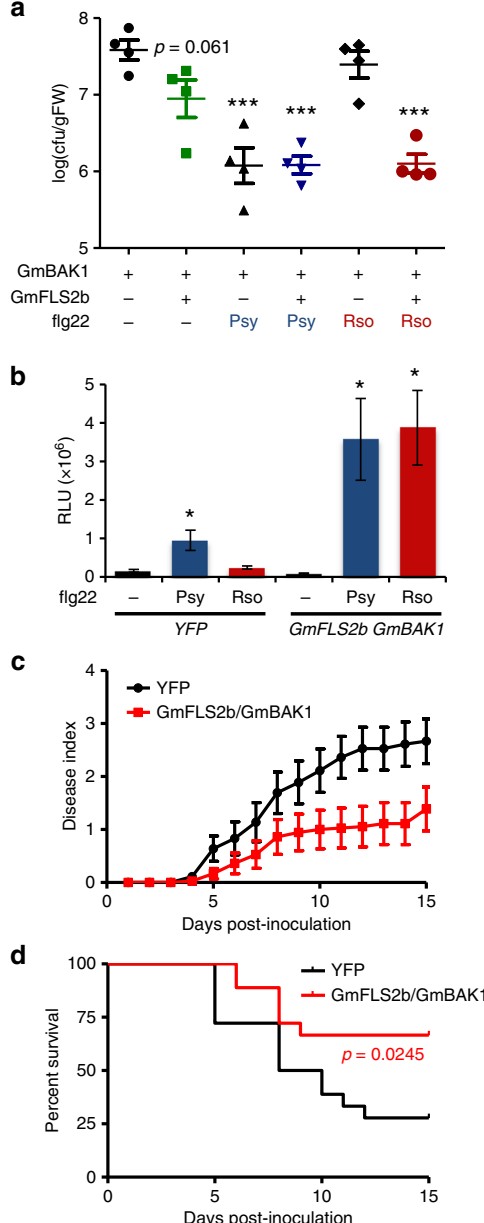

**Fig. 4 GmFLS2/GmBAK1 enhance resistance to *R. solanacearum*.**
**a** Growth of *R. solanacearum* Y45 in *N. benthamiana* leaves expressing GmBAK1 (as control) or GmFLS2b/GmBAK1, as indicated. Leaves were pretreated with water (mock), 1 µM flg22[Psy], or 1 µM flg22[Rso] for 12 h and then syringe-infiltrated with a 10^6 CFU mL^−1 inoculum. Bacterial growth was determined 24 h after inoculation (mean ± SEM, $n = 4$). Disease monitoring in subsequent days is shown in Supplementary Fig. 13. Asterisks indicate significant differences compared to the mock control expressing GmBAK1 as established by a Student's *t* test ($p < 0.001$). The *p* value is indicated for the reproducible attenuation observed in tissues expressing GmFLS2b/GmBAK1 with mock treatment. This experiment was performed three times with similar results. **b** ROS burst triggered by flg22[Psy] (100 nM), flg22[Rso] (100 nM) or a mock (water) treatment in tomato roots expressing YFP (control) or GmFLS2b/GmBAK1, measured in a luminol-based assay, and represented as accumulated relative luminescence units (RLU) (mean ± SEM, $n = 16$). The measurement was performed from 1 to 60 min after treatment. Asterisks indicate significant differences compared to the mock control as established by a Student's *t* test ($p < 0.05$). **c** Soil-drenching inoculation assays in tomato plants with roots expressing YFP (as control) or GmFLS2b/GmBAK1. Plants were inoculated with *R. solanacearum* GMI1000. The results are represented as disease progression, showing the average wilting symptoms in a scale from 0 to 4 (mean ± SEM, $n = 12$). **d** Survival analysis of tomato plants in (**c**). The disease scoring was transformed into binary data with the following criteria: a disease index lower than 2 was defined as '0', while a disease index equal or higher than 2 was defined as '1' for each specific time point. Statistical analysis was performed using a Log-rank (Mantel–Cox) test ($n = 12$) to analyse the difference with the control, and the resulting *p* value is indicated in the figure. Soil-drenching inoculation assays were performed four times, and representative results are shown here. Composite data for all the replicates are shown in Supplementary Fig. 14.

*hypogaea* cv. Hefeng 6, donated by Boshou Liao), medicago (*Medicago truncatula* A17, donated by Jian-Kang Zhu), and pea (*Pisum sativum* cv. Little Marvel) plants were grown under the same conditions as *N. benthamiana*. *Arabidopsis thaliana* plants were grown in a growth chamber under controlled conditions (22 °C under a 10-h light/14-h dark photoperiod with a light-intensity of 100–150 µE m^−2 s^−1).

For pathogen inoculation assays, tomato (*Solanum lycopersicum* cv. Moneymaker) and soybean plants were cultivated in jiffy pots (Jiffy International, Kristiansand, Norway) in a growth chamber under controlled conditions (25 °C with a 16 h light/8 h dark photoperiod, 130 µE m^−2 s^−1, 65% humidity) during 4 or 2 weeks, respectively. After soil-drenching inoculation, the plants were kept in a growth chamber under the following conditions: 27 °C with 12-h light/12-h dark photoperiod, a light-intensity of 130 µE m^−2 s^−1 and 75% humidity.

**Chemicals**. All the peptides used in this study were purchased from Abclonal (Wuhan, Hubei, China), and the sequences are detailed in the Supplementary Table 2. All other chemicals were purchased from Sigma-Aldrich unless otherwise stated.

**Plasmids, bacterial strains, and cultivation conditions**. *Ralstonia solanacearum* GMI1000 and Y45 were grown on solid BG medium[37] plates or cultivated overnight in liquid BG medium at 28 °C[38]. The *AtFLS2* (AT5G46330.2) gene was amplified using cDNA as template, cloned in pENTR-D-TOPO (ThermoFisher, Waltham, MA, USA), and subcloned into pGWB505[39] by LR reaction (ThermoFisher). The *GmFLS2a* (Glyma.08g083300.1), *GmBAK1* (Glyma.15G051600.1) (with stop codon), and *AtBAK1* (AT4G33430.2) (with stop codon) genes were amplified using cDNA as template, cloned in pDONR-207 (ThermoFisher), and subcloned into pGWB505 by LR reaction. The *GmFLS2b* (Glyma.05g128200.1) gene was amplified, cloned in pDONR-zeocin (ThermoFisher), and subcloned into pGWB505 by LR reaction. The fragments containing *GmFLS2a* and *GmFLS2b* were differentiated by sequencing. The *BAK1* genes from *Arabidopsis* and soybean without stop codon were cloned in pDONR-207 were also subcloned into pGWB514[39]. The plasmid co-expressing *GmFLS2* and *GmBAK1* was generated by Golden Gate technology as summarized in the Supplementary Fig. 15. DNA fragments (level 0 modules) were amplified from pDONR-207 (GmBAK1) or pDONR-zeocin (GmFLS2b) and assembled using *Bsa*I into level 1 vector pICH47751 (GmBAK1 promoter with GmBAK1 cds or YFP) or pICH47761 (35 S promoter with GmFLS2 cds or YFP). The final constructs comprising kanamycin resistance cassette, GmBAK1 and GmFLS2b or YFP were assembled using *Bpi*I into the binary vector

PRRs[15], which may have resulted in the gain of recognition of polymorphic PAMPs in certain plant species, as revealed by our results and those reported by Furst et al.[35]. Therefore, in addition to the transfer of new PRRs, the expression of additional PRR alleles, with extended ligand recognition capabilities, could contribute to the generation of broad-spectrum disease resistance in crops.

Our work suggests that the transfer of the GmFLS2/GmBAK1 pair to crop plants could represent a new strategy to enhance resistance to bacterial wilt disease. Moreover, unravelling the key residues responsible for both the evasion of perception by flg22[Rso] in Arabidopsis and the gain of perception by the soybean receptors paves the way for synthetic biology approaches to customize immune receptors to expand their range of recognition, enabling detection of polymorphic pathogen elicitors.

## Methods

**Plant materials and growth conditions**. *Nicotiana benthamiana* plants were grown on soil at one plant per pot in an environmentally-controlled growth room at 22 °C under a 16-h light/8-h dark photoperiod with a light-intensity of 100–150 µE m^−2 s^−1. Bean (*Phaseolus vulgaris* cv. Canadian Wonder), soybean (*Glycine max* cv. Williams82, donated by Suomeng Dong), peanut (*Arachis*

pAGM4723. The MoClo Toolkit[40] was a gift from Sylvester Marillonnet (Addgene kit #1000000044). Binary vectors were transformed into *Agrobacterium tumefaciens* (Agrobacterium) GV3101 for transient expression in *N. benthamiana* leaves or into *Agrobacterium rhizogenes MSU440* for expression in tomato roots (see details below). Agrobacterium carrying binary vectors was grown at 28 °C and 220 rpm in LB medium supplemented with rifampicin (50 mg L$^{-1}$), gentamycin (25 mg L$^{-1}$), and spectinomycin (50 mg L$^{-1}$).

**Site-directed mutagenesis.** GmFLS2b mutant variants were generated using the QuickChange Lightning Site-Directed Mutagenesis Kit (Life technologies, Waltham, MA, USA) following the manufacturer's instructions. The *GmFLS2b/pDONR-zeocin* plasmid was used as template. Primers used for the mutagenesis are listed in Supplementary Table 1.

**Agrobacterium-mediated gene expression in *N. benthamiana*.** Agrobacterium-mediated gene expression in *N. benthamiana* was performed as previously described[41]. Briefly, Agrobacterium carrying the different plasmids were suspended in infiltration buffer to a final OD$_{600}$ of 0.25 or 0.5 and infiltrated into the abaxial side of the leaves using a 1 mL needless syringe. Leaf samples were taken at 2 to 3 dpi (days post infiltration) for analysis based on experimental requirements.

**Protein extraction and western blots.** Protein extraction and western blots were performed as previously described[41]. Briefly, plant tissues were collected into 2 mL tubes with metal beads and frozen in liquid nitrogen. After grinding with a tissue lyser (Qiagen, Hilden, Germany) for 1 min at 25 rpm s$^{-1}$, proteins were extracted using protein extraction buffer (100 mM Tris-HCl, pH7.5; 10% glycerol; 2% NP40, 5 mM EDTA; 2 mM dithiothreitol; 1X proteinase inhibitor cocktail; 2 mM phenylmethylsulfonyl fluoride, 10 mM Na$_2$MoO$_4$, 10 mM NaF, 2 mM Na$_3$VO$_4$) and incubated for 10 min at 4 °C. After centrifugation (10 min; 15,000 × $g$), the supernatants were mixed with SDS loading buffer, denatured at 70 °C for 20 min, and resolved using SDS-PAGE. Proteins were transferred to a PVDF membrane and monitored by western blot using anti-GFP (Abicode, CA, USA, M0802-3a) and anti-HA (Roche, Basel, Switzerland, No.11583816001) antibodies. Both antibodies were diluted 1:5000.

**Measurement of ROS generation and MAPK activation.** PAMP-triggered ROS burst and MAPK activation in plant leaves were measured as described previously[42,43]. ROS in tomato roots was measured as described previously[13]. Briefly, plant tissues were placed in 96-well plates containing distilled water, and ROS was elicited with 100 nM flg22$^{Psy}$, flg22$^{Rso}$ or 100 nM of the indicated recombinant proteins. Luminescence was measured over 60 min using a microplate luminescence reader (Varioskan flash, Thermo Scientific, USA). MAPK activation assays were performed using 5-week-old *N. benthamiana* plants. Two days after Agrobacterium infiltration at OD$_{600}$ of 0.25, the intact leaves were elicited for 10 min after syringe infiltration of 1 μM flg22. Leaf discs were taken to monitor MAPK activation by western blot with anti-Phospho-p44/42 MAPK antibody (Erk1/2; Thr-202/Tyr-204, Cell Signaling, Danvers, MA, USA, 4370S). Anti-MAPK antibody was diluted 1:5000.

**Ralstonia solanacearum virulence assays.** *R. solanacearum* soil-drenching inoculation was performed as previously described[38] with specific modifications. Soybean or tomato plants grown in Jiffy pots for 12–14 and 21–28 days, respectively, were inoculated by drenching the soil with a bacterial suspension containing 10$^8$ colony-forming units per mL (CFU mL$^{-1}$). 35 mL of a suspension of *R. solanacearum* GM1000 was used to soak each pot. After incubation for 20 min with the bacterial inoculum, plants were transferred from the bacterial solution to a bed of potting mixture soil in a new tray[44]. Scoring of visual disease symptoms on the basis of a scale ranging from '0' (no symptoms) to '4' (complete wilting)[44]. To perform survival analysis, the disease scoring was transformed into binary data with the criteria: a disease index lower than 2 was defined as '0', while a disease index equal or higher than 2 was defined as '1' in terms of the corresponding time (days post-inoculation, dpi)[45].

Stem injection assays with *R. solanacearum* were performed as previously described[46]. Briefly, 10 μL of a 10$^6$ CFU mL$^{-1}$ bacterial suspension was injected into the stems of 4-week-old tomato plants and 2-week-old soybean plants. Injections were performed in the cotyledon emerging site in the stem, and photographs were taken at the indicated time points.

For *R. solanacearum* infiltration in *N. benthamiana* leaves, *GmFLS2/GmBAK1* or *GmBAK1* (as control) were expressed in the same *N. benthamiana* leaf using Agrobacterium with a final OD$_{600}$ of 0.25. PAMP-induced resistance assays were performed following the original procedure for Arabidopsis plants[33], with several modifications in our experimental system. Briefly, 24 h after Agrobacterium infiltration, water (as control) or a 1 μM solution of the indicated peptides was infiltrated into the leaves. Twelve hours after infiltration of the peptides, a 10$^6$ CFU mL$^{-1}$ inoculum of *R. solanacearum* Y45[34] was infiltrated into the same tissues. *R. solanacearum* Y45 was originally isolated from tobacco[47], and is able to replicate rapidly upon infiltration into *N. benthamiana* leaves[34]. This strain has been transformed with the pRCT-orange plasmid to confer resistance to tetracycline. Samples were taken 1 or 2 dpi to quantify CFU per gram of tissue. CFU were

counted by spreading serial dilutions on solid BG medium containing kanamycin (10 mg/l) to allow the selection of Y45 CFU.

**Co-immunoprecipitation.** Co-immunoprecipitation assays were performed as previously described[41] with several modifications. Briefly, *N. benthamiana* leaves were infiltrated with Agrobacterium containing pGWB505-*GmFLS2* (or mutant variants) and pGWB514-*GmBAK1* or pGWB505-*AtFLS2* and pGWB514-*AtBAK1*. Forty-eight hours after Agrobacterium infiltration, leaves were treated with water (as control), or a solution containing 1 μM of the indicated peptides for 10 min. Total proteins were extracted as indicated above and immunoprecipitation was performed with 15 μL of GFP-trap beads (ChromoTek, Munich, Germany) with 1-h incubation at 4 °C. Beads were washed 5 times with wash buffer with 1% NP40. The proteins were stripped from the beads by heating in 30 μL Laemmli buffer for 20 min at 70 °C. The immunoprecipitated proteins were separated on SDS-PAGE gels for western blot analysis with the indicated antibodies.

**Confocal microscopy.** Confocal imaging was performed as previously described[48]. Briefly, *GmFLS2-GFP* and mutant variants were expressed in *N. benthamiana* using Agrobacterium. Samples were imaged 2.5 days later on a Leica TCS SMD FLCS point scanning confocal microscope using the settings for visualizing GFP fluorescence with laser excitation: 488 nm, emission: 500–550 nm.

**FRET-FLIM.** Förster resonance energy transfer—fluorescence lifetime imaging (FRET-FLIM) experiments were performed as previously described[49,50] with several modifications. Briefly, GmFLS2b (fused to GFP) was expressed from pGWB505, and GmBAK1 (fused to RFP) or free RFP (as negative control) were expressed from pGWB554. FRET-FLIM experiments were performed on a Leica TCS SMD FLCS confocal microscope excitation with WLL (white light laser) and emission collected by a SMD SPAD (single photon-sensitive avalanche photodiodes) detector. Two days after infiltration, *N. benthamiana* plants transiently co-expressing donor and acceptor proteins were treated either with water or 1 μM flg22$^{Rso}$, as indicated in the figures, and visualized under the microscope 15–30 min after treatment. Accumulation of the GFP- and RFP-tagged proteins was estimated before measuring lifetime. The tuneable WLL set at 488 nm with a pulsed frequency of 40 MHz was used for excitation, and emission was detected using SMD GFP/RFP Filter Cube (with GFP: 500–550 nm). The fluorescence lifetime shown in the figures corresponding to the average fluorescence lifetime of the donor was collected and analysed by PicoQuant SymphoTime software. Lifetime is normally amplitude-weighted mean value using the data from the single (GFP-fused donor protein only or GFP-fused donor protein with free RFP acceptor or with non-interacting RFP-fused acceptor protein) or biexponential fit (GFP-fused donor protein interacting with RFP-fused acceptor protein). Mean lifetimes are presented as mean ± SEM based on eight images from three independent experiments.

**Tomato root transformation.** Tomato root transformation was performed using *Agrobacterium rhizogenes MSU440*[51]. Radicles of tomato seedlings were cut 7 days after germination, and the bottom of the hypocotyls were incubated with *Agrobacterium rhizogenes MSU440* carrying plasmids to express *GmFLS2/GmBAK1* and *YFP/YFP* (as control), and selected using plates of Murashige-Skoog medium containing kanamycin (50 mg L$^{-1}$). The responsiveness conferred by the expressed soybean genes was validated by determining the ROS burst triggered by flg22$^{Rso}$ (100 nM). Three weeks after transformation, seedlings were transferred to Jiffy pots, and soil-drenching inoculation with *R. solanacearum* (OD$_{600}$ of 0.01) was performed three-to-four weeks later as described above. Symptoms were scored as described above.

**Obtaining the initial models used in the structural analysis.** Initial atomic coordinates for the AtFLS2/AtBAK1/flg22$^{Pae}$ ternary complex were obtained from the published crystal structure of flg22 in complex with the FLS2 and BAK1 ectodomains[5] available from the Protein Data Bank under pdb ID 4mn8. The C-terminus alanine of flg22$^{Pae}$ was manually added and oriented based on visual inspection of the electron density data. To obtain coordinates for the AtFLS2/AtBAK1/flg22$^{Rso}$ complex, the 9 residue modifications were modelled using the FoldX computer algorithm[52]. Prior to any mutagenesis, the RepairPDB option of FoldX was used to optimize the total energy of the triplex, by identifying and repairing those residues that have bad torsion angles and van der Waals clashes. Mutagenesis was performed using the BuildModel option of FoldX, allowing for rotamer and sidechain orientation optimization of the mutated residues and neighbour residues both in the flg22 peptide and the receptor proteins. Finally, the initial atomic coordinates for the GmFLSb/GmBAK1/flg22$^{Rso}$ were obtained following a stepwise approach: primary sequence alignment with BLAST[53] of the GmFLS2b and GmBAK1 extracellular domains to the AtFLS2 and AtBAK1 sequence templates separately; based on the sequence alignments, putative 3D models for GmFLS2b and GmBAK1 were generated by comparative homology modelling with Swiss model;[54] the model structures were then structurally aligned with MUSTANG[55] to the equilibrated triplex structure of AtFLS2/AtBAK1/flg22$^{Rso}$ to obtain a complete set of atomic positions; clashed were removed using the RepairPDB option of FoldX.

**Model refinement by molecular dynamics equilibration**. All complex structures were further optimised by performing explicit solvent molecular dynamics simulations with periodic boundary conditions within a rectangular cell using the AMBER 18 suite of programs, with the pmemd.cuda module for GPUs[56], molecular interactions were represented with the ff14SB force field for proteins[57], Dang parameters for the ions[58] and SPC/E water[59]. The system was neutralized with 28 sodium ions placed at random within the simulation cell. The complex was then solvated with a layer of water at least 10 Å thick. Long-range electrostatic effects were treated using the Particle Mesh Ewald method with standard defaults[60], using a real-space cutoff of 10 Å. The length of chemical bonds involving hydrogen were restrained using SHAKE[61] and the Berendsen algorithm was used to control the temperature and the pressure[62], with a coupling constant of 5 ps. In order to equilibrate the complex side chains and solvent shell, a multistep protocol was followed, which involves energy minimizations of the solvent while keeping the solute rigid, slow thermalization and a final equilibration for 10 ns with position restraints on the backbone atoms. Visualization of states was conducted in PyMOL 2.3 molecular graphics system software, using PyMOL Tcl script language.

**Binding free energy estimations**. The change in binding free energy upon mutation from flg22$^{Pae}$ to flg22$^{Rso}$ was estimated using the BeAtMuSiC program[63] based on the structure of the protein–protein complex. Given that it has been suggested that the recognition of flg22 by the FLS2/BAK1 receptor complex takes place in two distinct steps[5], involving first the binding of flg22 to FLS2, followed by the recruitment of BAK1, binding free energy changes were calculated separately for the AtFLS2 + flg22 complex and for the triplex formation step. The latter refers to the binding energy of BAK1 to the already formed complex of AtFLS2/flg22. Alanine Scanning of the all interface residues in the complex was performed using the Robetta server[64]. The procedure identifies residues that are involved in the protein–protein interface, and uses a simple free energy function to calculate the changes in the binding free energy upon single substitutions of each side-chain to alanine. Here also, the two presumed stages of the recognition process were analysed separately, first the binding of flg22 to AtFLS2, then the binding of AtBAK1 to the complex formed by the other two. This analysis was interpreted to reveal hotspots in the structure where polymorphisms between plant species might have evolved for the recognition of the flg22$^{Rso}$.

**Phylogenetic analysis of flg22$^{Rso}$ sequences**. The flg22 sequences extracted from 155 sequenced *R. solanacearum* strains were kindly provided by Nemo Peeters. The name of the sequenced strains and the metadata associated to the genome sequences has been previously published[65].

**Recombinant protein purification from *Escherichia coli***. The recombinant GFP and full-length FliC (flagellin) used in this work were purified in a previous study[13]. Mass-spectrometry analysis of the recombinant proteins and other details about the purification process were published before[13].

**Statistical analysis**. Statistical analyses were performed with Prism 5 software (GraphPad). The data are presented as mean ± SEM. The statistical analysis methods are described in the figure legends.

**Reporting summary**. Further information on research design is available in the Nature Research Reporting Summary linked to this article.

## Data availability

The authors declare that all the data supporting the findings of this study are available within the paper and its Supplementary information files. Source data are provided with this paper.

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

## Acknowledgements

We thank Georg Felix for helpful discussions, Suomeng Dong, Jian-Kang Zhu, and Boshou Liao for sharing biological materials, Nemo Peeters for providing the alignment of flg22^Rso sequences, Rosa Lozano-Duran for critical reading of this paper, Haiyan Zhuang for technical assistance, and Xinyu Jian for technical and administrative assistance during this work. We thank Modesto Orozco for general supervision of structural analysis and access to advanced computing infrastructure. We thank the PSC Cell Biology core facility for assistance with confocal microscopy. This work was supported by the Strategic Priority Research Program of the Chinese Academy of Sciences (CAS) (grant XDB27040204), the National Natural Science Foundation of China (NSFC; grant 31571973), the Chinese 1000 Talents Program, and the Shanghai Center for Plant Stress Biology (CAS). JSR is funded by a President's International Fellowship Initiative (PIFI) postdoctoral fellowship (No. 2018PB0057 and 2020PB0088) from the Chinese Academy of Sciences. C.S. is supported by the Creative-Pioneering Researchers Program through Seoul National University. The CAS Center of Excellence in Molecular Plant Sciences has filed a patent application (pending) on behalf of inventors A.P.M. and Y.W. on the use of GmFLS2 and GmBAK1 to confer disease resistance in plants.

## Author contributions

Y.W. and A.P.M designed the work, supervised experiments, and analysed data. Y.W. performed most of the experimental work. A.B. performed homology-based structural modelling. J.S.R. assisted with cloning and performed FRET-FLIM analysis. C.S. generated the GmFLS2b/GmBAK1 co-expression construct. A.Z. and R.J.L.M. helped to perform gene expression in tomato roots followed by *R. solanacearum* inoculation. Y.W. and A.P.M. wrote the paper with inputs from all the authors.

## Competing interests

The authors declare no competing interests.
