## [Peer Review File · Nature Communications]

REVIEWER COMMENTS

Reviewer #1 (Remarks to the Author):

The manuscript "An immune receptor complex evolved in soybean to perceive a polymorphic bacterial flagellin" by Wei et. al. describes biologically meaningful diversification of an important microbe-associated molecular pattern and an important plant immunity-conferring pattern recognition receptor. The authors identify the mutations in flg22 from *Ralstonia solanacearum* that are responsible for that allele of flg22 not eliciting immune responses on most plants. The authors also identify that the orthologues of FLS2 in *Glycine max* are able to confer sensitivity to *R. solanacearum* flg22 when transiently expressed in *Nicotiana benthamiana* and *Solanum lycopersicum*. Through a series of elegant experiments, the authors identify several amino acid residues of Gm-FLS2 responsible for sensing *R. solanacearum* flg22. These results suggest that transferring of new alleles of plant pattern recognition receptors may confer improved disease resistance and that targeted mutagenesis of endogenous pattern recognition receptors may also improve plant disease resistance through enhanced detection of invading pathogens. The manuscript is very well written and expected to be of broad interest to the molecular plant-microbe interactions community. The experimental designs are rigorous with sufficient methodological details provided. The presented results encompass a very large amount of work. The conclusions are largely supported by the presented data. I outline my specific concerns with the manuscript below.

General comments:

Several of the conclusions are not fully supported without the supplementary information. I believe the paper would be improved by incorporating some of the supplementary information into the main text. As a starting point, I suggest that Figures S1f/g, S4, and S5 are moved to the main text.

Much previous research related to the results of this manuscript is not discussed or provided for context in this manuscript. A few examples: 1) GMFLS2 has already been described and shown to detect *Psy* flagellin (<https://doi.org/10.1016/j.plantsci.2019.110386>). 2) There are several known cases of polymorphisms in bacterial flg22 leading to reduced perception (for example Pma strain ES4326 flg22 also has mutations in the identified critical region of Rso flg22 and also does not trigger immunity in *Arabidopsis* (<https://doi.org/10.1111/nph.12408>)). 3) The concept of moving PRRs among crop families has been extensively considered in previous work. This should be better addressed in the text of this manuscript. This manuscript provides a great example of taking the concept further by discussing swapping alleles of 'more effective' PRRs rather than just introducing new PRRs. The idea of allelic diversification in PRRs is not new, however, and this manuscript needs to provide some context on that issue. For example, there is marked allelic diversification of *Arabidopsis* FLS2 (<https://doi.org/10.1093/molbev/mss011>). Other examples and discussion of allele swapping of PRRs are available in this review: <https://doi.org/10.1146/annurev-phyto-102313-045907>.

Specific comments:

Ln33: Delete "exceptionally"

Ln 77: Not accurate to say "Furthermore, whether any plants have evolved to recognize flg22Rso remains unknown" given the results presented in the manuscript.

Ln 124: The statement "and all of them show similar polymorphisms to the predominant flg22Rso sequence in residues 9, 18, 19, 20, and 21 (Figure S2)" is not accurate because 17% have an A to S polymorphism leading to potential strong differences in polarity.

One panel of Figure S3 is already a panel in Figure 2.

Figure S4: The descriptions of the "left" and "right" panels are reversed. The right panel is

unnecessary.

Figure 2B-D: Why was the ROS measurement started at 5 minutes, not 0 minutes like in the other figures? The initial high level of ROS due to accumulation overnight following wounding has been reported in many previous papers and is fine to show.

Figure 2E: Why is there an 'empty' lane?

Figure 2E: Concluding that GMFLS2b and GMBak1 interact specifically in the presence of flg22-Rso is not convincing based on the co-IP. There is only a faint band for the alpha-HA in the relevant column and the band for the crude HA is much stronger than the band for the crude HA in the column showing GMFLS2b and GMBAK1 without any flg22.

Supplementary information Ln121: Because infiltration of *Ralstonia* into tobacco leaves is not a typical inoculation method, can the authors provide some citations for when that method is employed? I'm not immediately sure if it is expected for *Ralstonia* to grow in tobacco leaves following infiltration. These results are also surprising because it's often observed that for plant protection assays (pretreating with immunity-eliciting peptides) to work effectively, spray or dip inoculation is required over infiltration inoculation.

Figure S7: It would be helpful to link each citation to each highlighted region of known importance for the protein interactions.

Figure 3: It is not clear in the text whether mutagenized variants were created for all of the locations highlighted in Figures 3a-3i. If so, why are only a subset of the results shown in the bottom half of Figure 3 and the related supplementary figures? If not, the sites that were not empirically tested through mutagenesis/transient expression should be removed from the figure or highlighted in some way to indicate that.

Figure S11A: Why is the localization of GMFLS2B showed repeatedly but several of the mutagenized variants are not shown at all?

Figure S11B: Protein accumulation levels are only shown for a subset of the tested mutagenized FLS2 variants. Protein accumulation needs to be shown for all. Non-differential responsiveness may be due to a mutation that does not affect binding/heterodimerization or due to differences in protein accumulation. In other words, all protein accumulations need to be shown to argue against false negatives.

Figure S11C: Reorder to be in the same order as panels of Figure 3.

Figure S12: Results for the Q248E transiently expressed variant should be included.

Ln231: Can the authors support the assertion that the differential responsiveness of the mutants is backed up by the mapK assay? It seems challenging to conclude that differential levels of mapK suggest a quantitative, differential amount of responsiveness given the uneven amounts of the proteins based on immunoblots of the anti-GFP antibody.

Figure 4A: What is the p-value for the reduction in the growth of *Ralstonia* on the N. bent leaves transiently expressing both GmFLS2 and GmBak1 but not pretreated with flg22 (the green dots)?

Figure S15: How is S15A different from Fig 4A?

Signed,
C. Clarke

Reviewer #2 (Remarks to the Author):

Review on submitted article Nature Communications
Manuscript number: NCOMMS-20-11709-T

Title: An immune receptor complex evolved in soybean to perceive a polymorphic bacterial flagellin

Authors: Yali Wei¹, Alexandra Balaceanu, Jose S. Rufian, Cecile Segonzac, Achen Zhao, Rafael J. L. Morcillo, and Alberto P. Macho

Overall evaluation:

The authors nicely investigated the molecular mechanism by which *Ralstonia solanacearum*, the causal agent of the bacterial wilt disease in many plant species, presents a polymorphic sequence in its flg22 epitope (flg22Rso) that avoids perception by most of plant species. Such polymorphisms abolish the recognition by the corresponding plant immune receptor complex FLS2/BAK1, notably in *Arabidopsis*.

First, the authors identified 9 amino acid polymorphisms between the active elicitor peptide from *Pseudomonas aeruginosa* flg22Pae and the sequence of flg22Rso. By in silico analysis and targeted mutations, the authors identified A21 as a crucial amino acid for the ability of flg22Rso to evade plant perception as the mutation I21A in the sequence of the flg22 epitope from *Pseudomonas syringae* (flg22Psy I21A) is sufficient to abolish the ROS burst normally induced in *Arabidopsis* and to trigger a detectable interaction between AtFLS2 and AtBAK1 revealed by Co-IP after transient expression in *Nicotiana benthamiana*.

Secondly, the authors identified an exception in the Fabaceae family showing that only the soybean (*Glycine max*) was able to naturally detect flg22Rso by inducing a ROS burst. The sequenced genome of soybean contains two FLS2 orthologs, named GmFLS2a and GmFLS2b, encoding proteins 54.47% and 53.69% identical to AtFLS2, that have been previously described (Tian et al., 2020). The expression of GmFLS2a or GmFLS2b and GmBAK1 in non-responsive plants confers full responsiveness to flg22Rso.

In the third part, the authors made an alanine scanning in the receptor complex to locate important residues that show mutations or structural discrepancies between the AtFLS2/AtBAK1/flg22Pae and GmFLS2b/GmBAK1/flg22Rso. By site-directed mutagenesis and transient expression in *N. benthamiana*, the authors showed that mutation of GmFLS2b-Q368 to F reduced the responsiveness to flg22Rso by approximately 50% compared to wild-type and that mutation of GmFLS2-R483 to I reduced responsiveness to flg22Rso by approximately 40%, indicating that the polymorphism in these 2 sites are determinant for the recognition of flg22Rso by GmFLS2b.

In the last part of the manuscript, the authors confirmed the role of GmFLS2b/GmBAK1 in the natural resistance of soybean to *Ralstonia solanacearum*. Then they expressed GmFLS2/GmBAK1 in tomato roots conferring responsiveness to flg22Rso and rendering tomato plants more resistant to disease upon soil-drenching inoculation with *R. solanacearum* GMI1000.

On the whole, the authors made an impressive and convincing work to understand in depth the molecular mechanisms of the evasion of a pathogenic bacterium and show how co-evolution permits to some plant species, such as soybean, to counterselect mutations to still detect the flagellin of *Ralstonia solanacearum*, providing a better natural resistance to this pathogen. If the concept that PRR gene transfer could confer plant resistance against pathogens has been shown before (Lacombe et al., 2010), this is one of the first really nice demonstration that inter-family transfer of immune receptor complex is a suitable strategy to enhance resistance in crops. Nevertheless I think that some questions detailed hereafter need to be answered in order to strengthen the quality of the manuscript before its publication.

Detailed evaluation:

Figure 1a: Why the sequence and the ROS production triggered by flg22Pae are not indicated whereas polymorphisms indicated in Fig.1b are between flg22Pae and flg22Rso? The reader cannot easily understand the studied mutations if the "reference" flg22 sequence is not the same in fig.1a and 1b! What is the "reference" flg22 sequence: flg22Psy or flg22Pae? Please also add a reference which indicates that *Pseudomonas aeruginosa* is a plant pathogen.

Figure 2a: Explain earlier in the legend why the measurement was performed from 5 to 40 min after treatment and not from 0 to 40 min?

Figure 2e: why two bands are detected why anti-HA in the crude extracts? What were the tested modalities in the second lane, strangely "empty"? The FRET-FLIM results shown in Figure S6 seem to be more convincing! Why not put them here?

Figure 3: why did the authors make the structure/function analysis on GmFLS2b whereas GmFLS2a is unable to interact with NbBAK1 and gave a higher ROS response (Fig. 2c, d)? The authors have to explain why they "choose" GmFLS2b instead of GmFLS2a, the most obvious to study! How the "representative results" have been chosen and why not replace them by Fig. S11c with at least n=5?

Figure S1: The experimental results of ROS production and CoIP are obtained with the WT version of flg22Psy or its mutated version flg22Psy I21A whereas the detailed structure shown is the one of flg22Pae! Why the authors did not show the 3D structure of flg22Psy? In Fig. S1c the mutation G18A doesn't seem to really modify the interactions as mentioned by the authors in the text.

Figure S3: the authors should explain earlier in the text why the results between time 0 and 5 min are not shown?

Figure S4: "FliC" is not explained. Is it the whole flagellin protein encoded by the fliC gene from *Ralstonia* and *Pseudomonas*? What is the concentration used?

Figure S5: why the authors used GmFLS2a only in this figure? Is it AtFLS2/AtBAK1? Fig. S5b: please indicate which promoter was used?

Figure S6: Why there is no statistical test indicated here? Are the red results significant? Is it GmFLS2a or 2b? Interaction and interact appear twice in the title, please simplify! Did you make the control AtFLS2 + AtBAK1 + flg22Rso?

Figure S7: Please highlight the key amino acids revealed by this study via alanine scanning and/or ROS production. Explain why the last T is underlined in green?

Figure S9: Please explain the red color code. Try to change the flg22 color which mixed up with the PRR sequences.

Figure S11b: where is the western blot for Q248E? What is the loading control? Ponceau red staining or anti actin?

Minor corrections:

Summary:

Please try to shorten the summary and suppress any reference to agree with the instructions for authors.

Text part:

- L.123 : pos.21 = S or A so please not use "all of them"
- L.159 : « which we named GmFLS2a and GmFLS2b » -> these 2 genes have been studied and named before so please add the corresponding reference (Tian et al., 2020)
- L.169 : « (Figure 2d and S5). » please suppress S5 because not shown
- L.189: the authors claimed that alignments of sequences from GmFLS2 and GmBAK1 with their orthologues in *Arabidopsis* did not permit to identify significant polymorphisms in key residues previously shown to interact with flg22 whereas some are present such as R152>G in FLS2.

Figures :

- Fig.1b : please add in the table : "Binding Free energy loss"
- Fig.2c&d : add "b" after GmFLS2 "GmFLS2b-GFP"
- Fig.3 : "5-7 times" replace by "5-6 times" as shown in fig.S11

- Fig.S10 : please add « b » to GmFLS2 "GmFLS2b/flg22Rso/GmBAK1"

Mat. &Met. :

- Protein extraction and western blots :

« β -mercapitoethanol » -> β -mercaptoethanol

- *Ralstonia solanacearum* virulence assays

"for twelve-to-fourteen and twenty one-to-twenty" -> suppress "-"

"20-minute" -> 20 minutes

Please find below a point-by-point response to the reviewers' comments (our responses are in bold blue text). We are submitting a revised manuscript with tracked changes and a "clean" version, with all the changes accepted, to help revision (please note that the line numbers mentioned in this response correspond to the "clean" version).

Sincerely,
Alberto Macho (on behalf of all authors)

REVIEWER COMMENTS

Reviewer #1 (Remarks to the Author):

The manuscript "An immune receptor complex evolved in soybean to perceive a polymorphic bacterial flagellin" by Wei et. al. describes biologically meaningful diversification of an important microbe-associated molecular pattern and an important plant immunity-conferring pattern recognition receptor. The authors identify the mutations in flg22 from *Ralstonia solanacearum* that are responsible for that allele of flg22 not eliciting immune responses on most plants. The authors also identify that the orthologues of FLS2 in *Glycine max* are able to confer sensitivity to *R. solanacearum* flg22 when transiently expressed in *Nicotiana benthamiana* and *Solanum lycopersicum*. Through a series of elegant experiments, the authors identify several amino acid residues of Gm-FLS2 responsible for sensing *R. solanacearum* flg22. These results suggest that transferring of new alleles of plant pattern recognition receptors may confer improved disease resistance and that targeted mutagenesis of endogenous pattern recognition receptors may also improve plant disease resistance through enhanced detection of invading pathogens. The manuscript is very well written and expected to be of broad interest to the molecular plant-microbe

interactions community. The experimental designs are rigorous with sufficient methodological details provided. The presented results encompass a very large amount of work. The conclusions are largely supported by the presented data. I outline my specific concerns with the manuscript below.

General comments:

Several of the conclusions are not fully supported without the supplementary information. I believe the paper would be improved by incorporating some of the supplementary information into the main text. As a starting point, I suggest that Figures S1f/g, S4, and S5 are moved to the main text.

>>Response: We thank the reviewer for this suggestion. Most of the shortage regarding figure panels, text, and references was due to the limitations in the original submission to Nature, which was directly transferred to Nature Communications. Following the reviewer's advice, we have now incorporated several supporting figures to the main figures:

- Figure S1f/g are now Figure 1e/f.

- Figure S4 is now Figure 2c.

- Figure S5: we have reorganized the data to show the ROS curves and protein accumulation as Figure 2d and 2e. We keep the Arabidopsis controls as Figure S4, which is now better described in the text.

- Figure S6 is now Figure 2g.

Much previous research related to the results of this manuscript is not discussed or provided for context in this manuscript. A few examples: 1) GMFLS2 has already been described and shown to detect Psy flagellin (<https://doi.org/10.1016/j.plantsci.2019.110386>).

>>Response: We thank the reviewer for pointing this out. We have provided additional context in this manuscript by mentioning, citing, and discussing previous research. Regarding this first example, we have cited the corresponding reference about the GmFLS2 genes and discussed the results from that paper in the last section of our manuscript (lines 230-231).

2) There are several known cases of polymorphisms in bacterial flg22 leading to reduced perception (for example Pma strain ES4326 flg22 also has mutations in the

identified critical region of Rso flg22 and also does not trigger immunity in Arabidopsis (<https://doi.org/10.1111/nph.12408>)).

>>Response: We have now included in the introduction the reference to the polymorphic flg22 in other bacterial pathogens, including ES4326 and Agrobacterium, and the consideration that allelic diversification in PAMPs represents a suitable pathogen virulence strategy to avoid perception by plants (Vinatzer et al, 2014) (lines 95-104).

3) The concept of moving PRRs among crop families has been extensively considered in previous work. This should be better addressed in the text of this manuscript. This manuscript provides a great example of taking the concept further by discussing swapping alleles of 'more effective' PRRs rather than just introducing new PRRs. The idea of allelic diversification in PRRs is not new, however, and this manuscript needs to provide some context on that issue. For example, there is marked allelic diversification of Arabidopsis FLS2 (<https://doi.org/10.1093/molbev/mss011>). Other examples and discussion of allele swapping of PRRS are available in this review: <https://doi.org/10.1146/annurev-phyto-102313-045907>.

>>Response: We thank the reviewer for these suggestions. We have now indicated in the text that the transfer of PRRs between different plant species has been extensively used to confer additional responsiveness to pathogen elicitors, enhancing plant resistance to the corresponding pathogens, and cited several examples where this approach was used (Afroz et al, 2011; Albert et al, 2015; Du et al, 2015; Lacombe et al, 2010; Lu et al, 2015; Schwessinger et al, 2015; Schoonbeek et al, 2015; Tripathi et al, 2014; Mendes et al, 2010; Fradin et al, 2011; Pfeilmeier et al, 2019) (lines 356-361).

We also discuss now the allelic diversification of plant PRRs, citing the interesting discussion provided in the suggested review (Vinatzer et al, 2014). It is possible that such diversification may have resulted in the gain of recognition of polymorphic PAMPs, and our results (and those reported by Furst et al, 2020) support this notion. Therefore, we now included additional text to suggest that, in addition to the transfer of new PRRs, the expression of additional PRR alleles with extended ligand recognition capabilities could contribute to the generation of broad-spectrum disease resistance in crops (lines 406-414).

Specific comments:

Ln33: Delete “exceptionally”

>>Response: We have deleted “exceptionally”.

Ln 77: Not accurate to say “Furthermore, whether any plants have evolved to recognize flg22R_{so} remains unknown” given the results presented in the manuscript.

>>Response: We have removed that sentence.

Ln 124: The statement “and all of them show similar polymorphisms to the predominant flg22R_{so} sequence in residues 9, 18, 19, 20, and 21 (Figure S2)” is not accurate because 17% have an A to S polymorphism leading to potential strong differences in polarity.

>>Response: We thank the reviewer for pointing this out. We have changed the sentence to make it more accurate, and it now states that all the strains show the same polymorphisms as the predominant flg22^{R_{so}} sequence in residues 9, 18, 19, and 20 (Figure S2); in the residue 21, a 16.3% of the strains presented an S, instead of the predominant A present in most strains (83.3%) (Figure S2).

One panel of Figure S3 is already a panel in Figure 2.

>>Response: That figure panel was provided for comparison, but we have now removed the panel in Figure S3 to avoid duplications.

Figure S4: The descriptions of the “left” and “right” panels are reversed. The right panel is unnecessary.

>>Response: We thank the reviewer for pointing this out. The left panel has now been moved to Figure 2c, and the right panel has been removed.

Figure 2B-D: Why was the ROS measurement started at 5 minutes, not 0 minutes like in the other figures? The initial high level of ROS due to accumulation overnight following wounding has been reported in many previous papers and is fine to show.

>>Response: We now provide the whole curve starting from 0 minutes in these panels.

Figure 2E: Why is there an 'empty' lane?

>>Response: We always observed a stronger signal upon flg22^{Psy} treatment, particularly in MAPK activation. Still, we thought that it was interesting to show it in the same blot for comparison. Then, to avoid interference of this signal with the weaker signal in other lanes, we left an empty lane in the blot. We have included a column in the legend (-) to avoid confusion and indicate that this lane is indeed empty.

Figure 2E: Concluding that GMFLS2b and GMBak1 interact specifically in the presence of flg22-Rso is not convincing based on the co-IP. There is only a faint band for the alpha-HA in the relevant column and the band for the crude HA is much stronger than the band for the crude HA in the column showing GMFLS2b and GMBAK1 without any flg22.

>>Response: We thank the reviewer for pointing this out. We always observed flg22^{Rso}-dependent interaction in different replicates, and we have now replaced the blots for a different replicate, where the protein amounts in the crude extract are even.

Supplementary information Ln121: Because infiltration of *Ralstonia* into tobacco leaves is not a typical inoculation method, can the authors provide some citations for when that method is employed? I'm not immediately sure if it is expected for *Ralstonia* to grow in tobacco leaves following infiltration. These results are also surprising because it's often observed that for plant protection assays (pretreating with immunity-eliciting peptides) to work effectively, spray or dip inoculation is required over infiltration inoculation.

>>Response: The reviewer is right that many *Ralstonia* strains would not grow in *N. benthamiana* leaves, since 2 T3Es present in these strains (AvrA and PopP1) would trigger immunity. However, for these assays, we used the Y45 strain, which was originally isolated from tobacco (Li et al, 2011: <https://www.ncbi.nlm.nih.gov/pmc/articles/PMC3194909/>), and lacks both AvrA and PopP1 (<https://iant.toulouse.inra.fr/bacteria/annotation/site/prj/T3Ev3>). We

have recently shown that, upon low dose infiltration into *N. benthamiana* leaves, Y45 can replicate rapidly to 10^7 - 10^8 CFU/g of tissue (see Figure 2B in Sang et al, 2020: <https://doi.org/10.1016/j.xplc.2020.100025>); in Sang et al (2020) we also show that Y45 growth is restricted by the prior activation of immunity in the infiltrated tissues. We have now clarified this by adding a brief mention in the main text (line 367) and a detailed explanation in the methods section (lines 556-557).

Regarding the plant protection assays (PAMP-induced resistance), the original protocol described in Zipfel et al (2004) uses infiltration of the PAMP followed by infiltration of the bacteria in the leaves. The same procedure has been used in many subsequent papers that analyse PAMP-induced resistance. We have always used infiltration-infiltration for PAMPs and bacteria, with satisfactory results regarding plant protection. We have now cited the original protocol before explaining our specific procedure in the methods section (lines 550-552).

Figure S7: It would be helpful to link each citation to each highlighted region of known importance for the protein interactions.

>>Response: We thank the reviewer for this suggestion. We have now included the citation for each highlighted region in the new Figure S5.

Figure 3: It is not clear in the text whether mutagenized variants were created for all of the locations highlighted in Figures 3a-3i. If so, why are only a subset of the results shown in the bottom half of Figure 3 and the related supplementary figures? If not, the sites that were not empirically tested through mutagenesis/transient expression should be removed from the figure or highlighted in some way to indicate that.

>>Response: We apologize if there was a lack of clarity in the text. Indeed, mutagenized variants were created for all the locations highlighted in these figure panels. Please note that the position of the residues Q248, Q368, N 391, R483, and T507 in GmFLS2 is equivalent to that of the residues E247, F369, H392, I483, and I507, respectively, in AtFLS2, as shown in Figure S5. All the mutants were tested and shown in the bottom half of Figure 3. We have now clarified this in the legend of Figure 3 in the revised manuscript version.

Figure S11A: Why is the localization of GMFLS2B showed repeatedly but several of the mutagenized variants are not shown at all?

>>Response: Since *N. benthamiana* assays may display leaf-to-leaf variation, mutant variants are shown next to their respective wild-type control expressed side-by-side in the same leaf, which represents the same setup used for the ROS assays shown in Figure 3. We have now updated the figure and the figure legend to clarify this.

Figure S11B: Protein accumulation levels are only shown for a subset of the tested mutagenized FLS2 variants. Protein accumulation needs to be shown for all. Non-differential responsiveness may be due to a mutation that does not affect binding/heterodimerization or due to differences in protein accumulation. In other words, all protein accumulations need to be shown to argue against false negatives.

>>Response: We now show protein accumulation levels for all the mutagenized FLS2 variants in the new Figure S9. The figure shows that the negative result obtained for the Q248E mutant is not due to differences in protein accumulation (see confocal image in S9a and western blot in S9b).

Figure S11C: Reorder to be in the same order as panels of Figure 3.

>>Response: All the panels in this figure have been reordered accordingly in the new Figure S9.

Figure S12: Results for the Q248E transiently expressed variant should be included.

>>Response: These results are now shown (new Figure S10).

Ln231: Can the authors support the assertion that the differential responsiveness of the mutants is backed up by the mapK assay? It seems challenging to conclude that differential levels of mapK suggest a quantitative, differential amount of responsiveness given the uneven amounts of the proteins based on immunoblots of the anti-GFP antibody.

>>Response: We have now replaced that figure for a different replicate with similar GmFLS2-GFP accumulation for all the mutants (new Figure S10), which shows that the reduced responsiveness of the Q368F and the R483I mutants is backed up by the pMAPK assay. The manuscript text has been slightly edited to mention this specifically (lines 342-343).

Figure 4A: What is the p-value for the reduction in the growth of *Ralstonia* on the N. bent leaves transiently expressing both GmFLS2 and GmBak1 but not pretreated with flg22 (the green dots)?

>>Response: We now indicate the p value for this sample (0.061). We thank the reviewer for pointing this out; as we mention in the text, this attenuation was always reproducible, but never statistically significant. We have now included the p value in the figure to reflect that it is indeed very close to the usual threshold of 0.05.

Figure S15: How is S15A different from Fig 4A?

>>Response: Figure 4a showed the bacterial numbers 24 hours post-inoculation, while the old Figure S15a showed the numbers 2 days post-inoculation. A sentence has been added to the legend of Figure 4a to indicate that disease monitoring in subsequent days is shown in Figure S13.

Signed,
C. Clarke

Reviewer 2

Remarks to author:

Overall evaluation:

The authors nicely investigated the molecular mechanism by which *Ralstonia solanacearum*, the causal agent of the bacterial wilt disease in many plant species, presents a polymorphic sequence in its flg22 epitope (flg22Rso) that avoids perception by most of plant species. Such polymorphisms abolish the recognition by the corresponding plant immune receptor complex FLS2/BAK1, notably in *Arabidopsis*.

First, the authors identified 9 amino acid polymorphisms between the active elicitor peptide from *Pseudomonas aeruginosa* flg22Pae and the sequence of flg22Rso. By in silico analysis and targeted mutations, the authors identified A21 as a crucial amino acid for the ability of flg22Rso to evade plant perception as the mutation I21A in the sequence of the flg22 epitope from *Pseudomonas syringae* (flg22Psy I21A) is sufficient to abolish the ROS burst normally induced in *Arabidopsis* and to trigger a detectable interaction between AtFLS2 and AtBAK1 revealed by Co-IP after transient expression in *Nicotiana benthamiana*.

Secondly, the authors identified an exception in the Fabaceae family showing that only the soybean (*Glycine max*) was able to naturally detect flg22Rso by inducing a ROS burst. The sequenced genome of soybean contains two FLS2 orthologs, named GmFLS2a and GmFLS2b, encoding proteins 54.47% and 53.69% identical to AtFLS2, that have been previously described (Tian et al., 2020). The expression of GmFLS2a or GmFLS2b and GmBAK1 in non-responsive plants confers full responsiveness to flg22Rso.

In the third part, the authors made an alanine scanning in the receptor complex to locate important residues that show mutations or structural discrepancies between the AtFLS2/AtBAK1/flg22Pae and GmFLS2b/GmBAK1/flg22Rso. By site-directed mutagenesis and transient expression in *N. benthamiana*, the authors showed that mutation of GmFLS2b-Q368 to F reduced the responsiveness to flg22Rso by approximately 50% compared to wild-type and that mutation of GmFLS2b-R483 to I reduced responsiveness to flg22Rso by approximately 40%, indicating that the polymorphism in these 2 sites are determinant for the recognition of flg22Rso by GmFLS2b.

In the last part of the manuscript, the authors confirmed the role of GmFLS2b/GmBAK1 in the natural resistance of soybean to *Ralstonia solanacearum*. Then they expressed GmFLS2/GmBAK1 in tomato roots conferring responsiveness

to flg22Rso and rendering tomato plants more resistant to disease upon soil-drenching inoculation with *R. solanacearum* GMI1000.

On the whole, the authors made an impressive and convincing work to understand in depth the molecular mechanisms of the evasion of a pathogenic bacterium and show how co-evolution permits to some plant species, such as soybean, to counterselect mutations to still detect the flagellin of *Ralstonia solanacearum*, providing a better natural resistance to this pathogen. If the concept that PRR gene transfer could confer plant resistance against pathogens has been shown before (Lacombe et al., 2010), this is one of the first really nice demonstration that inter-family transfer of immune receptor complex is a suitable strategy to enhance resistance in crops. Nevertheless I think that some questions detailed hereafter need to be answered in order to strengthen the quality of the manuscript before its publication.

Detailed evaluation:

Figure 1a: Why the sequence and the ROS production triggered by flg22Pae are not indicated whereas polymorphisms indicated in Fig.1b are between flg22Pae and flg22Rso? The reader cannot easily understand the studied mutations if the “reference” flg22 sequence is not the same in fig.1a and 1b! What is the “reference” flg22 sequence: flg22Psy or flg22Pae? Please also add a reference which indicates that *Pseudomonas aeruginosa* is a plant pathogen.

>>Response: As positive control for functional assays, we used flg22 from *Pseudomonas syringae* (flg22^{Psy}), which is a well-characterized plant pathogen. All the modelling work associated to this study was done with flg22 from *P. aeruginosa* (flg22^{Pae}), since we used the structural data published using this peptide (Sun et al, 2013), as indicated in the methods section. Please note that flg22 peptides from different *Pseudomonas* species, including *P. aeruginosa* and *P. syringae* display high similarity, showing identical sequence in the residues 9-22 (Figure S1), which comprise the key residues analysed by mutagenesis in Figure 1a. To help the readers and improve clarity, we have included this explanation in the main text (lines 135-137) and in the legend of Figure S1. We have also included a reference of *P. aeruginosa* as a plant pathogen (lines 126-128).

Figure 2a: Explain earlier in the legend why the measurement was performed from 5 to 40 min after treatment and not from 0 to 40 min?

>>Response: In the revised version of the figure, we show the measurements from 0 to 40 minutes.

Figure 2e: why two bands are detected why anti-HA in the crude extracts? What were the tested modalities in the second lane, strangely “empty”?

>>Response: The two bands detected with anti-HA in crude extracts appeared in some of our replicates of this assay, but were not reproducible. We supposed that the higher band could correspond to phosphorylated BAK1 upon PAMP treatment, which may only be detected in specific favourable conditions. However, given that such higher band did not show in all the replicates (and that we do not study BAK1 phosphorylation in this work), we did not discuss this further. In fact, given the suggestion by Reviewer 1, we now show a different replicate with equal protein amounts in the crude extract, and such second band is not visible in this replicate.

Regarding the empty lane, we have included a column in the legend (-) to avoid confusion and indicate that this lane is indeed empty. We do this to avoid interference of the signal of the first lane with the weaker signal in other lanes (see also response to Reviewer 1).

The FRET-FLIM results shown in Figure S6 seem to be more convincing! Why not put them here?

>>Response: We thank the reviewer for this suggestion. We have now moved the FRET-FLIM results to the main Figure 2.

Figure 3: why did the authors make the structure/function analysis on GmFLS2b whereas GmFLS2a is unable to interact with NbBAK1 and gave a higher ROS response (Fig. 2c, d)? The authors have to explain why they “choose” GmFLS2b instead of GmFLS2a, the most obvious to study!

>>Response: We apologize for the lack of clarity in this regard. The absolute values in ROS assays are highly variable among replicates, and that is why we

always compare constructs side-by-side in the same assay. When comparing GmFLS2a with GmFLS2b side-by-side, GmFLS2b always gave a stronger response, which may not be appreciated in our previous representation of independent assays. The new Figure 2d and 2e show a side-by-side comparison, where it can be appreciated that GmFLS2b is more efficient in responding to flg22^{Rso}, and this is now explained in the text (lines 243-246). Considering this stronger responsiveness, we decided to use GmFLS2b for the structural modelling analysis, and a sentence to point this out has been included in the main text (lines 286-287).

How the “representative results” have been chosen and why not replace them by Fig. S11c with at least n=5?

>>Response: The results shown in Figure 3 were chosen as representative of the values obtained in, at least, 5 independent biological repeats based on similarity to the average of all the independent repeats. However, ROS assays upon transient expression in *N. benthamiana* are notoriously variable, and therefore we provide the values from all the repeats, as well as their average difference to the control, in Figure S11 (now Figure S9). A representative result is shown in detail (including the ROS dynamics) in the main figure, while the results from all the repeats are provided as supporting materials for the sake of transparency.

Figure S1: The experimental results of ROS production and ColP are obtained with the WT version of flg22^{Psy} or its mutated version flg22^{Psy} I21A whereas the detailed structure shown is the one of flg22^{Pae}! Why the authors did not show the 3D structure of flg22^{Psy}? In Fig. S1c the mutation G18A doesn't seem to really modify the interactions as mentioned by the authors in the text.

>>Response: The crystal structure of Arabidopsis FLS2 and BAK1 ectodomains has been solved in complex with flg22^{Pae} (Sun et al, 2013) (a sentence has been added to the text to clarify this; lines 126-128), and therefore we did not have structural data to use flg22^{Psy} as a template for structural modelling. Nevertheless, please note that, as mentioned above, flg22 peptides from different *Pseudomonas* species, including *P. aeruginosa* and *P. syringae* display high similarity, showing identical sequence in the residues 9-22 (Figure S1), which comprise the key residues analysed by mutagenesis in

Figure 1a. To help the readers and improve clarity, we have included this explanation in the main text (lines 135-137) and in the legend of Figure S1.

Regarding the G18A mutation, the reviewer is right in that it probably does not change the interactions with BAK1 residues. As described before (Sun et al, 2013), mutations in that residue would generate steric clashes with the BAK1 loop and consequently attenuate their interaction, rather than modifying or creating new interactions. To make this clear, the manuscript text now states: “For the second step, in keeping with previously published results, a mutation in G18 to A is predicted to compromise binding affinity (Figure 1b and S1c), probably by causing steric clashes that attenuate the existing interactions with BAK1 residues (Sun et al, 2013).” (lines 168-171).

Figure S3: the authors should explain earlier in the text why the results between time 0 and 5 min are not shown?

>>Response: We now show all the data from 0-40 minutes.

Figure S4: “FliC” is not explained. Is it the whole flagellin protein encoded by the fliC gene from *Ralstonia* and *Pseudomonas*? What is the concentration used?

>>Response: We have modified the figure legend (now in Figure 2) to clarify that FliC corresponds to the full-length recombinant flagellin. We have also specified the concentration used (100 nM) in the figure legend.

Figure S5: why the authors used GmFLS2a only in this figure? Is it AtFLS2/AtBAK1?

>>Response: we have re-arranged these figures. The old figure S5 (now S4) is used to show that the overexpression of Arabidopsis PRRs does not confer responsiveness to flg22^{Rso}. This is now mentioned more clearly in the manuscript (lines 243-246).

Fig. S5b: please indicate which promoter was used?

>>Response: we used 35S promoter. This is now indicated in the manuscript text and in the figure legend of Figures 2 and S4.

Figure S6: Why there is no statistical test indicated here? Are the red results significant? Is it GmFLS2a or 2b? Interaction and interact appear twice in the title, please simplify! Did you make the control AtFLS2 + AtBAK1 + flg22Rso?

>>Response: We have now moved this figure to the main Figure 2, as suggested by the reviewer, and have included statistical analysis to show that the red results are statistically significant. We have also added a label to indicate that we used GmFLS2b. We did not include a control using AtFLS2/AtBAK1, since we never observed any elicitation activity of flg22Rso in *N. benthamiana* expressing these receptors (Figure S4), and we used a non-elicited sample as negative control.

Figure S7: Please highlight the key amino acids revealed by this study via alanine scanning and/or ROS production. Explain why the last T is underlined in green?

>>Response: We thank the reviewer for this suggestion. We have now highlighted with green boxes the key amino acids in our analysis (new Figure S5). The underline in the last T has been removed.

Figure S9: Please explain the red color code. Try to change the flg22 color which mixed up with the PRR sequences.

>>Response: We have added an explanation for the color code, which was used to represent changes in binding free energy, and changed the flg22 color to purple to differentiate it from PRRs (new Figure S7).

Figure S11b: where is the western blot for Q248E ? What is the loading control? Ponceau red staining or anti actin?

>>Response: We now show protein accumulation levels for all the mutagenized FLS2 variants in the new Figure S9b. Anti-actin was used as loading control for all the western blots.

Minor corrections:

Summary:

Please try to shorten the summary and suppress any reference to agree with the instructions for authors.

>>Response: We have shortened the summary and removed the references.

Text part:

- L.123 : pos.21 = S or A so please not use "all of them"

>>Response: We have changed the sentence to make it more accurate, and it now states that all the strains show the same polymorphisms as the predominant flg22^{Rso} sequence in residues 9, 18, 19, and 20 (Figure S2); in the residue 21, a 16.3% of the strains presented an S, instead of the predominant A present in most strains (83.3%) (Figure S2).

- L.159 : « which we named GmFLS2a and GmFLS2b » -> these 2 genes have been studied and named before so please add the corresponding reference (Tian et al., 2020)

>>Response: We thank the reviewer for pointing this out. We have cited the corresponding reference regarding the GmFLS2 genes and discussed the results from that paper in the last section of our manuscript (lines 416-417).

- L.169 : « (Figure 2d and S5). » please suppress S5 because not shown

>>Response: We have re-arranged these figures and the corresponding citations in the text.

- L.189: the authors claimed that alignments of sequences from GmFLS2 and GmBAK1 with their orthologues in Arabidopsis did not permit to identify significant polymorphisms in key residues previously shown to interact with flg22 whereas some are present such as R152>G in FLS2.

>>Response: We apologize for the inaccuracy of our previous statement. Indeed, R152 is important for the interaction with flg22 Q1, although this residue was not relevant for the evasion of perception by flg22^{Rso}. Therefore, we have rephrased that sentence to specify that the alignment did not identify significant polymorphisms in key residues known to mediate interaction with

flg22 residues that present relevant polymorphisms in flg22^{Rso} (lines 281-282).

Figures :

- Fig.1b : please add in the table : "Binding Free energy loss"
- Fig.2c&d : add "b" after GmFLS2 "GmFLS2b-GFP"
- Fig.3 : "5-7 times" replace by "5-6 times" as shown in fig.S11

>>Response: Corrected, thanks.

- Fig.S10 : please add « b » to GmFLS2 "GmFLS2b/flg22Rso/GmBAK1"

>>Response: Corrected, thanks.

Mat. &Met.:

- Protein extraction and western blots :
« β -mercapitoethanol » -> β -mercaptoethanol
- Ralstonia solanacearum virulence assays
"for twelve-to-fourteen and twenty one-to-twenty" -> suppress "-"
"20-minute" -> 20 minutes

>>Response: Corrected, thanks.

(please see attached file from reviewer #2 with formatting preserved)

REVIEWERS' COMMENTS:

Reviewer #1 (Remarks to the Author):

The authors have addressed all concerns noted in the previous review. The manuscript is excellent and expected to be of broad interest to the molecular plant-microbe interactions research community.

Reviewer #2 (Remarks to the Author):

On the whole, the quality of the manuscript has been greatly improved and most of the reviewers remarks have been taken into account.

After a new reading, I just see some minor things to change:

1/ scientifically, I think that 2 things need to be clarified:

- according to the amino-acids numbering showed in Figure S5, the mutation Q248 in Glycine max corresponds to E249 in Arabidopsis and not E247. As a consequence, E247 in Fig. 3 a and b and its legend must be renumbered as E249.
- in Fig. 2e, the reader expects to see an immunoblot detection to confirm the presence of GmBAK1 in the lanes GmBAK1-HA +. Either the authors show this additional detection of GmBAK1 with an anti-HA antibody, either they have to present this figure differently to agree with the text part where the authors only focus on the quantity of protein FLS2a or b, respectively.

2/ Concerning the form of this manuscript some discrepancies are still present concerning the presentation of some figures or legends. Thus, I would advise to:

- homogenize the legends concerning figures showing ROS dynamics (from 1 to 40 min) or ROS bar charts (integrated between 5 to 40 min to avoid the background noise between 0 and 5 min) using the legend of fig. 3 as a model.
- homogenize the presentation of the 3D structures shown in fig. 1, 3, S1 and S7 by labelling all the images by taking fig.1 and 3 as models (for example modify S1 by labelling all the 3-D structures with S1b, c, d, e, f, g, the corresponding legend and the text part).
- in figure 2g, please add -RFP and -GFP after GmBAK1 and GmFLS2b, respectively, because the FRET-FLIM is based on the fluorescence of these 2 proteins after the energy transfer.
- in Fig. S6, the mutation N->T is not easy to find so if you can highlight this mutation, it can improve the reading of the MS.
- in the legend of fig. S14, please complete the sentence ... of the experiment shown in figure 4c by adding "and d".
- in the figure S15, please add Gm before BAK1 cds in the level 2 of the construct (green part).

Benoit Poinsot

Please find below a point-by-point response to the reviewers' comments (our responses are in bold blue text).

Sincerely,

Alberto Macho (on behalf of all authors)

REVIEWERS' COMMENTS:

Reviewer #1 (Remarks to the Author):

The authors have addressed all concerns noted in the previous review. The manuscript is excellent and expected to be of broad interest to the molecular plant-microbe interactions research community.

Reviewer #2 (Remarks to the Author):

On the whole, the quality of the manuscript has been greatly improved and most of the reviewers remarks have been taken into account.

After a new reading, I just see some minor things to change:

1/ scientifically, I think that 2 things need to be clarified:

- according to the amino-acids numbering showed in Figure S5, the mutation Q248 in Glycine max corresponds to E249 in Arabidopsis and not E247. As a consequence, E247 in Fig. 3 a and b and its legend must be renumbered as E249.

>>Response: We thank the reviewer for pointing this out. We have corrected the figure and figure legend accordingly.

- in Fig. 2e, the reader expects to see an immunoblot detection to confirm the presence of GmBAK1 in the lanes GmBAK1-HA +. Either the authors show this additional detection of GmBAK1 with an anti-HA antibody, either they have to present this figure differently to agree with the text part where the authors only focus on the quantity of protein FLS2a or b, respectively.

>>Response: We thank the reviewer for pointing this out. We apologize, since there was an error in the labeling of the figure: as explained in the figure legend, these blots correspond to the protein accumulation in the assays shown in Figure 2d, which use untagged GmBAK1 (and not GmBAK1-HA). Untagged GmBAK1 cannot be detected by immunoblot, and that figure is indeed aimed to show only the accumulation of GmFLS2a/b. The indication of (-/+) for GmBAK1 is only provided to differentiate the specific samples used for ROS assays (-/+ GmBAK1) in Figure 2d. We have corrected the labeling of the figure, and the figure legend specifies that these samples include untagged GmBAK1 (in the explanation of Figure 2d).

2/ Concerning the form of this manuscript some discrepancies are still present concerning the presentation of some figures or legends. Thus, I would advise to:

- homogenize the legends concerning figures showing ROS dynamics (from 1 to 40 min) or ROS bar charts (integrated between 5 to 40 min to avoid the background noise between 0 and 5 min) using the legend of fig. 3 as a model.

>>Response: We thank the reviewer for this suggestion. We have homogenized the information included in each figure legend specifying the time covered by each representation.

- homogenize the presentation of the 3D structures shown in fig. 1, 3, S1 and S7 by labelling all the images by taking fig.1 and 3 as models (for example modify S1 by labelling all the 3-D structures with S1b, c, d, e, f, g, the corresponding legend and the text part).

>>Response: We thank the reviewer for this suggestion. We have homogenized the presentation of the 3D structures using Figure 1 and 3 as models.

- in figure 2g, please add -RFP and -GFP after GmBAK1 and GmFLS2b, respectively, because the FRET-FLIM is based on the fluorescence of these 2 proteins after the energy transfer.

>>Response: We thank the reviewer for this suggestion. The figure has been modified accordingly.

- in Fig. S6, the mutation N->T is not easy to find so if you can highlight this mutation, it can improve the reading of the MS.

>>Response: We thank the reviewer for this suggestion. The figure has been modified accordingly.

- in the legend of fig. S14, please complete the sentence ... of the experiment shown in figure 4c by adding "and d".

- in the figure S15, please add Gm before BAK1 cds in the level 2 of the construct (green part).

>>Response: We thank the reviewer for this suggestion. We have done the suggested modifications.

Benoit Poinssot